# SYNTHETIC DATASETS FOR MACHINE LEARNING ON SPATIO-TEMPORAL GRAPHS USING PDES

## ABSTRACT

In this work, we describe the creation and use of synthetic datasets based on various partial differential equations to support spatio-temporal graph modeling in machine learning for different applications. More precisely, we showcase three equations to model different types of disasters and hazards in the fields of epidemiology, atmospheric particles, and tsunami waves. Further, we show how such created datasets can be used by benchmarking several machine learning models on the epidemiological dataset and, additionally, by showing how pre-training on such synthetic datasets can improve model performance on real-world epidemiological data. The presented methods enable others to create datasets and benchmarks customized to individual requirements. The source code for our methodology and the three created datasets can be found on github.com/github-usr-ano/Temporal_Graph_Data_PDEs.

## 1 INTRODUCTION

Spatio-temporal graph modeling in machine learning has recently seen some developments (Chen & Eldardiry, 2024). Despite the existence of few regularly used data sources, we identify a lack of larger high-quality datasets for different tasks. A noteworthy source of data for spatio-temporal graph machine learning is traffic data from the *California PeMS system* (Li et al., 2018; Yu et al., 2018; Cini et al., 2023) in different settings. Also, COVID-19 data has been used in e.g. theoretical work classifying different architectures (Gao & Ribeiro, 2022). Although the existent datasets are valuable for specific tasks, they have limitations in quality (high noise), scope (few samples), accessibility (individually curated but unpublished), and adaptability (high individual preprocessing effort). These limitations hinder the comprehensive benchmarking and development of spatio-temporal graph architectures for different tasks. Therefore, large and more diverse high-quality datasets that are freely accessible are crucial.

The utilization of synthetic data allows the fast creation and adaption of datasets while avoiding barriers regarding privacy or data availability. Furthermore, they are less error-prone regarding measurement flaws and easier to control, as all involved dynamics are known in contrast to real-world data. Some examples of (partially) synthetic datasets that find application in machine learning are weather datasets (Muñoz Sabater et al., 2021), which are extensively used, along with turbulence data Li et al. (2008), PDE-data Takamoto et al. (2022) or ground motion data (Lehmann et al., 2024).

Numerous scientific processes can be described with partial differential equations (PDEs). While in many fields alternative modeling approaches exist, the versatility of PDE-based modeling is outstanding. Over the past centuries, considerable research efforts have been dedicated to advancing mathematical modeling with PDEs. Through our publication, we intend to make these significant developments accessible to the spatio-temporal graph learning community. Given their omnipresence in countless applications, we outline a method for generating synthetic datasets based on PDEs in this paper. Even though there exists a significant amount of work on PDEs and their solutions (Evans, 2022), the practical implementation of a PDE solver remains a demanding task.

Throughout this work, we solve three exemplary PDEs related to different types of disasters, each describing the movement of entities over space and time. We solve these PDEs on an irregular domain, and evaluate them on irregularly distributed points on this domain, as this is the realistic scenario for the presented applications. Subsequently, we form time-dependent graphs from the obtained values in combination with the underlying spatial structure, to which we grant researchers

direct access. While some prominent ML research focuses on the study of PDEs itself, such as Raissi et al. (2019); Li et al. (2021) where an analytic description of the dynamics has to be known, we hereby intend to mainly address data-centric studies of spatio-temporal graphs. Notably, we facilitate the adoption of our code to other domains, dynamics, PDEs, or even applications. Our work distinguishes itself from prior work (Takamoto et al., 2022) not only through the selected PDEs, but also the fundamental methodology, the the Finite Element Method (FEM). Specifically the FEM allows for complex domains with complex boundary conditions, and our overall outcome is a temporal graph, not a grid. We further emphasize the interchangeability of the (complex) domain, dynamics, or underlying equation throughout the utilization of the FEM and our published code.

The first equation we employ is inspired by the structure of real-world epidemiological data (e.g. COVID-19 incidences), which suggests the use of spatio-temporal graph machine learning (Nguyen et al., 2023; Gao & Ribeiro, 2022). We create a comparably large graph dataset containing long records of infectious data with different epidemiological settings but consistent measurements over time, absence of noise, and unconstrained accessibility to all determining information. This or any similar epidemiological PDE, based on the SIR-ODE (Kermack & McKendrick, 1991), has never been solved numerically. Especially, there are no accessible simulations for other practitioners. Furthermore, we want to enable epidemiological researchers, to use individual spatial domains (e.g. country) by adapting our published code to specifically train and test models on other geometries. Additionally, a broad collection of similar PDEs with slight adaptations can easily extend the modeling by features such as vaccination status (Schlickeiser & Kröger, 2021), exposed (He et al., 2020), or disease vectors (Collins & Duffy, 2022). Naturally, similar adaptations can also be applied to the following two equations.

Another prominent simulation technique in epidemiology for the spatio-temporal spread of infectious diseases are Agent-Based models (ABMs). AMBs often incorporate the SIR-ODE (Kermack & McKendrick, 1991) and simulate the spatial spread through the actions of agents, where an agent represents an individual entity, such as a person or animal, which stochastically interact with each other and their environment. While ABMs are commonly used to model macroscopical results of microscopical actions (Ferguson et al., 2006), they depend on many epidemiological assumptions, are stochastical and are limited to epidemiology. Modeling the spatial spread of a pathogen through a diffusion process via PDEs is much easier and more stable regarding skewed assumptions. We further illustrate our method of data-generation, alongside two more PDEs. Their resulting datasets are less applied to real-world problems but help researchers in development, exploration and benchmarking of new models.

The second equation, an advection-diffusion (or convection-diffusion) equation, models the movement of particles (e.g. dust, nuclear fallout, smoke) or other quantities in the atmosphere. The arrangement as a spatio-temporal graph as the underlying datatype is canonical, as real-life measurements of those quantities with a sensor network likewise have an irregular geometry. Besides a further benchmarking of temporal graphs, the creation of such a dataset could be used to pre-train either risk-prediction models or to pre-train or test specific parts of ML-based weather models such as ClimODE (Verma et al., 2024).

As a third exemplary PDE, we choose the wave equation, which is very common in physics and describes, for example, water, sound, or electromagnetism. In our case, we model a tsunami wave approaching some coastline, where an interesting task could be to predict the next few steps of the wave.

To demonstrate the utility of synthetic data, we lastly present a benchmarking of prominent spatio-temporal prediction models on the epidemiological data. Alongside, we test the expressivity-based classification concept of Gao & Ribeiro (2022), to understand whether it is sufficient to estimate the performance of different architectures and their quality in practice. We further underline the importance of synthetic data by evaluating if a transfer of knowledge into real-world data shows any benefit.

## 2 TEMPORAL PDEs TO SIMULATE SPATIO-TEMPORAL MOVEMENT

To simulate the proposed propagations through space and time through the three exemplary PDEs, we first generally outline the numerical approximation of the solution. In Section 3, we describe the generations of the datasets more explicitly.

We solve the PDEs on (different) 2D-domains $\Omega \subset \mathbb{R}^2$ for the three proposed scenarios. The PDEs are time-dependent on an interval $[0, T]$, and their solutions $\nu$ are of dimension $d \in \{1, 2\}$, depending on the equation. Generally, our PDEs have the form $\frac{\partial \nu}{\partial t} = F(\nu, \nabla \nu, \Delta \nu)$ with some additional boundary conditions. $F$ is some (general) functional which will be specified later, $\Delta$ the Laplace operator and $\nabla$ the Gradient, both only along the spatial dimensions.

The solution $\nu : [0, T] \times \Omega \rightarrow \mathbb{R}^d$ will be evaluated on a set of points within the domain $\Omega$. These points mimic geographical regions and their administrative units, or any unevenly distributed network of sensors. We equip this set of points with adjacencies and inverse distances to create a spatio-temporal graph dataset.

**Numerical solution sketch**  In the following paragraphs, we describe the numerical approximation of different time-dependent PDEs. This part can be skipped by a reader looking for a spatio-temporal graph dataset but does not want to bother with the details of its generation. However, to understand and adapt the published code and the data generation process, a short sketch of the methods is inevitable to us and important to a reader looking to expand our set of equations and parameters. Note, that although the outlined methodology can be applied to countless processes and PDEs, the complexity of the synthesized data is limited by the amount and complexity of given dynamics.

We solve the equations iteratively in the time domain and solve the occurring PDE for each time-step with the FEM, this order is called Rothe method. We therefore first discretize alongside the temporal dimension with an Euler scheme, i.e. approximate the time-derivative with a difference quotient. We use the Crank-Nicolson method, which is a mixture of the implicit and explicit Euler scheme for time-stepping. With $\theta = 0.5$, a small temporal Euler stepsize $h$, and $\nu = \nu(x, t)$ and $\nu_+ = \nu(x, t + h)$ the approximation has the form

$$\frac{\nu_+ - \nu}{h} = \theta F(\nu, \nabla \nu, \Delta \nu) + (1 - \theta) F(\nu_+, \nabla \nu_+, \Delta \nu_+).$$

Since for the timestep $t$ the solution $\nu$ is known, we thereby obtain a PDE that is not time-dependent anymore. We solve this PDE for $\nu^+$ with the FEM, which is the most flexible, reliable, and therefore standard numerical approach on PDEs. We only give a sketch of the FEM here, as details can be found in numerical textbooks for PDEs. The FEM discretizes the domain $\Omega$ into a mesh, on which a set of test functions $\phi$ as well as trial functions $\tilde{\nu}$ (approximation of the solution $\nu$) are defined. Throughout this work, we use first-order (linear) Lagrange Elements as test and trial functions. To obtain the so-called weak formulation, we rearrange the PDE into an implicit and an explicit part, then multiply it from the left with the test functions and integrate over the full domain. This forms an inner product, which allows an underlying Sobolev space to have the structure of a Hilbert space. We will not dive deeper into this, but use the notation of an inner product $\langle \phi, \nu \rangle = \int_\Omega \phi \nu \, dx$ for a better readability.

Replacing additionally $\nu, \nu_+$ with the corresponding trial functions $\tilde{\nu}, \tilde{\nu}_+$, we obtain the formulation:

$$\langle \phi, \tilde{\nu}_+ \rangle - h(1 - \theta)\langle \phi, F(\tilde{\nu}_+, \nabla \tilde{\nu}_+, \Delta \tilde{\nu}_+) \rangle = \langle \phi, \tilde{\nu} \rangle + h\theta \langle \phi, F(\tilde{\nu}, \nabla \tilde{\nu}, \Delta \tilde{\nu}) \rangle.$$

The left-hand-side (implicit side) can be written as $A(\tilde{\nu}_+)$, the right hand (explicit side) as $b$. We then seek to solve $A(\tilde{\nu}_+) = b$ for (the parameters of) $\tilde{\nu}_+$. Note, that technically $\phi$ is a vector of test functions, but we wanted to keep the notation simple.

One often employed identity for compact $\phi$ is $\langle \phi, \Delta \nu \rangle = -\langle \nabla \phi, \nabla \nu \rangle$ due to integration by parts, which we apply when Dirichlet or zero-flow Neumann boundary values are imposed. Note that we omit plenty of information on which space this inner product is defined and details of such transformations for the sake of simplicity, but again refer to any textbook covering the FEM.

### 2.1 SI-DIFFUSION EQUATION

To model the spatio-temporal spread of infectious diseases, we consider the following PDE from Murray (2003)

$$\frac{\partial S}{\partial t} = -rIS + D\Delta S,$$
$$\frac{\partial I}{\partial t} = rIS - \alpha I + D\Delta I, \tag{1}$$

where $\Delta$ denotes the Laplace operator and the functions $S(x,t), I(x,t)$ describe the densities of the susceptible and infected population over space and time.

The functions $\alpha(x,t), r(x,t)$, and $D(x,t)$ in the equation represent dynamics rising from pathogens and the population. More precisely, $r$ describes the transmission rate of the disease, $\alpha$ describes the duration of the disease, and $D$ describes the speed of the diffusion, i.e. movement of the population. Note that by setting $D = 0$ one receives the underlying SI compartment ODE, while setting $r = \alpha = 0$ leads to two time-dependent heat equations. Many adaptions of the underlying SI compartment ODE (He et al., 2020; Collins & Duffy, 2022) can easily be integrated into our framework.

As boundary condition, we chose mixed boundary conditions: usually, we impose zero-flow Neumann boundary conditions. However, to start a wave of infections, we impose small positive Dirichlet boundary conditions for a small part of the boundary for a limited time. Since this PDE is a system of two equations, the weak formulation utilizes two test functions $\phi_1, \phi_2$ from the same set of functions. The left-hand-side of the weak formulation, named $A$ above, here takes the form

$$A(\tilde{S}_+, \tilde{I}_+) = \langle \phi_1, \tilde{S}_+ \rangle + (1-\theta)h_k\langle \phi_1, r\tilde{S}_+\tilde{I}_+ \rangle + (1-\theta)h_k\langle \nabla\phi_1, +D\nabla S_+ \rangle$$
$$+ \langle \phi_2, \tilde{I}_+ \rangle - (1-\theta)h_k\langle \phi_2, r\tilde{S}_+\tilde{I}_+ - \alpha\tilde{I}_+ \rangle + (1-\theta)h_k\langle \nabla\phi_2, D\nabla\tilde{I}_+ \rangle$$

while the right-hand-side $b$ is defined by the following

$$b = \langle \phi_1, \tilde{S} \rangle - \theta h_k\langle \phi_1, r\tilde{S}\tilde{I} \rangle - \theta h_k\langle \nabla\phi_1, +D\nabla\tilde{S} \rangle$$
$$+ \langle \phi_2, \tilde{I} \rangle + \theta h_k\langle \phi_2, r\tilde{S}\tilde{I} - \alpha\tilde{I} \rangle - \theta h_k\langle \nabla\phi_2, D\nabla\tilde{I} \rangle.$$

Due to the product $\tilde{S}_+\tilde{I}_+$, the equation $A(\tilde{S}_+, \tilde{I}_+) = b$ depends nonlinearly on (the parameters of) the trial functions $\tilde{S}_+, \tilde{I}_+$. To solve this equation for (the parameters of) $\tilde{S}_+, \tilde{I}_+$, the Newton method is a standard choice, which we used. The Newton method solves this system iteratively, i.e. starts with $\tilde{S}_{+,1}, \tilde{I}_{+,1}$ and calculates $\tilde{S}_{+,2}, \tilde{I}_{+,2}$ until a convergence criterion is reached. The updates $\delta_S, \delta_I$ to $\tilde{S}_{+,i}, \tilde{I}_{+,i}$ are calculated as the solution of the linear system $J_A(\tilde{S}_{+,i}, \tilde{I}_{+,i})\delta = -A(\tilde{S}_{+,i}, \tilde{I}_{+,i}) + b$, where $J_A$ is the Jacobian matrix of $A(\tilde{S}_{+,i}, \tilde{I}_{+,i})$. The differentiation to set up $J_A$ is handled automatically.

## 2.2 Advection-diffusion equation

The Advection-diffusion equation takes the form

$$\frac{\partial u}{\partial t} = -\beta \cdot \nabla u + \alpha\Delta u + s, \tag{2}$$

with $u = u(x,t) \in \mathbb{R}$ being a measurement of the quantities density and the vector field $\beta = \beta(x,t) \in \mathbb{R}^2$ being the velocity field of e.g. wind over our domain. The diffusion coefficient $\alpha = \alpha(x,t)$ controls the diffusive spread of $u$ and $s = s(x,t)$ is a source term, with $s_+ = s(x, t+h)$.

As boundary condition, we assume Dirichlet boundary conditions $u|_{\partial\Omega} = 0$.

The explicit part, using integration by parts, takes the form:

$$b = \langle \phi, \tilde{u} \rangle + h\theta\langle \phi, \beta \cdot \nabla\tilde{u} \rangle + h\theta\alpha\langle \nabla\phi, \nabla\tilde{u} \rangle - h\theta\langle \phi, s \rangle - h(1-\theta)\langle \phi, s_+ \rangle$$

and the implicit part

$$A(\tilde{u}_+) = \langle \phi, \tilde{u}_+ \rangle - h(1-\theta)\langle \phi, \beta \cdot \nabla\tilde{u}_+ \rangle - h(1-\theta)\alpha\langle \nabla\phi, \nabla\tilde{u}_+ \rangle$$

that is linear in (the parameters of) $\tilde{u}_+$, such that we can solve a linear system

$$A\tilde{u}_+ = b.$$

## 2.3 WAVE EQUATION

The wave equation is a prototype time-dependent PDE, in which we added a damping term to damp simulated waves over time. The wave equation we will use takes the following form

$$\frac{\partial^2 u}{\partial t^2} + b\frac{\partial u}{\partial t} = \Delta u, \quad u(x, 0) = 0. \tag{3}$$

On the boundary of the domain we impose Robin boundary conditions $\alpha u + \beta \eta \cdot \nabla u = 0$, but with a local initial disturbance to start a wave that mimics an initial tsunami-wave. Due to the second derivative in the time domain, the equation doesn't suit our opening definition, but can easily be extended into a suitable system of PDEs:

$$\frac{\partial u}{\partial t} = v, \quad \frac{\partial v}{\partial t} + b\frac{\partial u}{\partial t} = \Delta u.$$

By replacing the time-derivatives, using the trial functions $\tilde{u}, \tilde{v}$ in the first equation, we obtain an implicit and explicit side again

$$(1 - (1-\theta)^2 h^2 \Delta + h(1-\Theta)b)\tilde{u}_+ = (1 + (1-\Theta)hb + \theta(1-\theta)h^2 \Delta\tilde{u} + h\tilde{v}$$
$$\tilde{v}_+ = h\Delta((1-\theta)\tilde{u}_+ + \theta\tilde{u}) + b\tilde{u} - b\tilde{u}_+ + \tilde{v}+,$$

from which the weak formulation can be derived easily. We solve the first equation (linear) for (the parameters of) $\tilde{u}_+$, and receive (the parameters of) $\tilde{v}_+$ straightforward from the second equation.[1]

## 3 DATASET CREATION

To compute the solutions of the three presented PDEs, we generate a mesh of the domain $\Omega$ with the meshing software *gmsh* (Geuzaine & Remacle, 2009). The implementation of the FEM is done with *deal.ii* (Arndt et al., 2023), which is written in *C++* and abstracted the full implementation around the FEM, but this can be done with any other FEM library.

As an underlying domain for both the epidemiological PDE and the advection-diffusion PDE, we use the shape of Germany. We evaluate the solutions of the PDEs on a set of points that have the coordinates of administrative regions in Germany: 400 *NUTS-3* regions. The intuition is the reporting of COVID-19 cases in the same spatial granularity. We additionally manually create an adjacency list of neighboring regions and weight them with the inverse of the distance of the centers of the regions, More details on this can be found in the appendix A.1. We ultimately have created a graph

$$\mathcal{G} = ((V, E, X_i))_{i=1,\dots,N}$$

with $400 = |V|$ nodes, $2088 = |E|$ edges and edge features $X_i \in \mathbb{R}^d$ for every time-step $i$. The mesh for the FEM computation and the resulting graph can be seen in Figure 1. On our GitHub we provide further explanations on how to increase the size of the graph $V$ with minor effort.

---

[1]The methodology we use here for an undamped ($b = 0$) wave equation, as well as similar numerical code, can be found in this tutorial of the used library https://www.dealii.org/current/doxygen/deal.II/step_23.html. We made only smaller adaptions to the code.

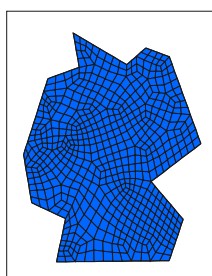 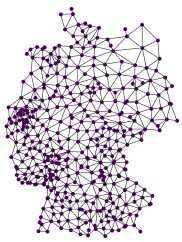 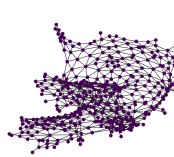

Figure 1: The mesh on the domain for the SI-diffusion equation and the Advection-diffusion equation (left). Independent of this mesh, we build a Graph using administrative regions (*NUTS-3*), their adjacencies, and the distances of the centers (center). Note, that we used the underlying geo-coordinates to plot this graph (center), while the later used and constructed graph has no trivial unique representation (right).

For the wave equation, the domain $\Omega$ is an imaginary (similar to the German) coastline, and the set of points on which we evaluate the solution of the wave equation are chosen randomly. Their respective adjacencies are selected with a Delaunay triangulation, and equipped with inverse distances to form a graph $\mathcal{G}$ with $325 = |V|$ nodes and $1858 = |E|$ edges. The mesh and the resulting graph can be seen in Figure 2.

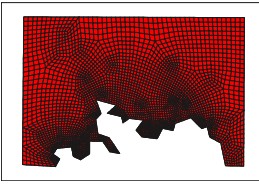 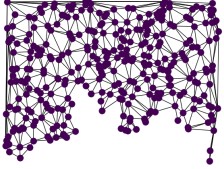

Figure 2: The mesh on the domain for the Wave Equation. On the same domain we generate a set of 325 random points and generate a Delaunay triangulation.

**SI-diffusion equation**  We set the parameter $\alpha = 0.22$, the time-weighting parameter $\Theta = 0.5$ effectively resulting in a Crank-Nicolson scheme, the Euler stepsize $h_k = 1$, and a Newton stepsize $\eta = 0.3$. The underlying domain $\Omega$ can be seen in Figure 1.

We generated 25 scenarios, each with a length of 364 time-steps. For every simulation we varied the parameters $r, D$, they can be found in the supplementary material in A.1 and Table 3, leading to a dataset of shape $[9100, 400, 2]$ for [time-steps, nodes, features] in 5:27 hours (wall clock) of computation. Note, that two features exist due to $2 = d = |\{S, I\}|$.

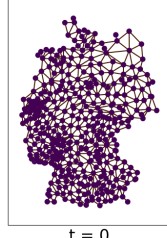 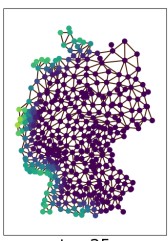 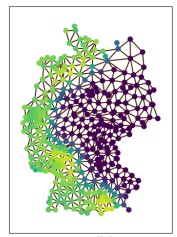 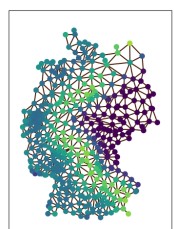 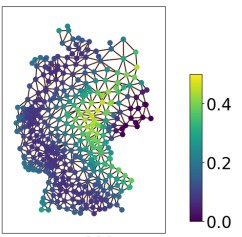

Figure 3: Plot of one scenario of an infection from the diffusion SI-Equation. The Infection wave propagates over the nodes.

**Advection-diffusion equation**  For the Advection-diffusion equation, we simulated 54 different parameters right after one another, each with a length of 80 time-steps. Throughout the simulation, we keep $\alpha = -1$ constant and also change $\beta(x, t)$ only every 80 time-steps. For the source term $s(x, t)$ we define a small constant support $\Omega_{s(x,t)} \subset \Omega$ for $s$, such that

$$s(x, t) = \begin{cases} S(t) & x \in \Omega_{s(x,t)} \\ 0 & x \notin \Omega_{s(x,t)} \end{cases}$$

where $\Omega_{s(x,t)}$ is a rectangle within the domain. $S(t)$ is a sine function. TODO Jost The choice of $h, T, S(t), \beta(x, t), \Omega_{s(x,t)}$ and further details can be found in the supplementary material A.1.2 Table 4, resulting in a dataset of shape $[4320, 400, 1]$, for [time-steps, nodes, features] in 0:22h. A visualization can be found in 7.

**Wave equation**  For the wave equation, we simulated one consecutive simulation in which we simulated two tsunami waves from different starting points with an initial amplitude of 1. We set $T = 2700000$ and $h = 100000/64$, resulting in a dataset of shape $[1858, 325, 1]$, for [time-steps, nodes, features] in 0:20h. Details can be found in the appendix A.1.3.

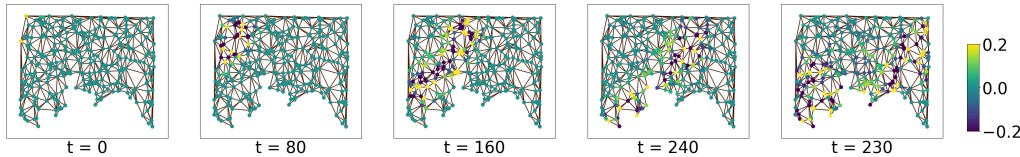

Figure 4: Plot of one scenario of an infection from the Wave-equation.

**Real-world epidemiological datasets**  Additionally to the synthetic datasets we created above, we want to compare the epidemiological dataset to real-world data. To do we create three real world datasets: German COVID-19, German Influenza, and Brazilian COVID-19. The numbers of infected people were taken from public sources. The German numbers were connected with the graph $\mathcal{G}$ from the SI-diffusion equation, for the Brazilian data we created a new graph $\mathcal{G}$ analogously by using geographic data of the regions. Details can be found in the appendix, section A.1.4.

## 4  APPLICATION OF SYNTHETIC DATA

To showcase the utility of our created datasets, we conduct several machine learning experiments in this section. We first define several interesting (spatio-) temporal machine learning architectures in Section 4.1. In section 4.2 we define three different benchmarking scenarios, to compare the performance of the previously defined machine learning models. To demonstrate that the usage of our data exceeds pure theoretical studies, we additionally perform experiments of transfer-learning from our synthetic data onto three real-world datasets from epidemiology in section 4.3.

### 4.1  (SPATIO-) TEMPORAL MODELS

We want to test solely data-centric models and do not take any knowledge of the underlying PDE into account, such as its structure or parameters. We test both temporal and spatio-temporal models. More specific details can be found in the supplementary material A.2 and the published material online.

**Repetition**  The repetition model is a naive baseline that simply repeats the last given value.

**RNN**  Recurrent-Neural Networks based on GRUs (Cho et al., 2014) have proven to be successful for sequence prediction. During training, teacher forcing is applied. During validation forecasts are produced autoregressively.

**TST**  Time Series Transformer (TST) is a Transformer-based (Vaswani et al., 2023) architecture designed for temporal forecasting. It utilizes encoder and decoder layers to capture temporal dependencies, and a positional encoding depending purely on the time-step. During training, teacher forcing is applied. During validation forecasts are produced autoregressively.

**MP-PDE** Given that the underlying data is based on solutions of a PDE, we also test a model based on a Message-Passing PDE-solver (MP-PDE) from Brandstetter et al. (2022), but will not pass any information of the underlying PDE to our model. The model consists of an encoder, a processor, and a decoder. The encoder creates node-wise embeddings of the context data. The processor consists of an MP-GNN, operating on a single graph with embedded features. The decoder is a 1D convolution applied node-wise, and a special update rule, that propagates the last value through the next time-step.

**RNN-GNN-Fusion** Motivated by the PDE itself, we aimed to separate the time from the space dimensions by building an RNN (analogously to the abovementioned model) to encode the underlying ODE and an MP-GNN to emulate the diffusion. RNN and GNN run parallel, their outputs are combined in a convex combination. During training, teacher-forcing is applied. During validation forecasts are produced autoregressively. In the classification of Gao & Ribeiro (2022) this would be classified as *time-and-graph*.

**GraphEncoding** We recreated a model from a recent contribution from the field of ML-based epidemiological forecasting (Nguyen et al., 2023). This model encodes the contextual time-steps separately in a shared GNN. The encoded graphs are then propagated through an LSTM (Hochreiter & Schmidhuber, 1997) network. To achieve a forecast for multiple days, this network is applied autoregressively. In the classification of Gao & Ribeiro (2022) this would be classified as *graph-then-time*.

## 4.2 EPIDEMIOLOGICAL BENCHMARKING

To benchmark the presented architectures from section 4.1, we define three tasks on the synthetic epidemiological dataset, created from the SI Eq. 1, that are motivated by forecasting tasks with real-world data. While we wanted to present thorough experiments on this dataset, results with a similar experimental setup, but on the other two datasets from Eq. 2 and Eq. 3, can be found in the appendix A.2.3.

We will proceed with the epidemiological dataset which we split $80/20$ along the time axis into a train and a test dataset, and omit any information about the Susceptible $S$, as this reflects real-world data. On this data, we define the following tasks for machine learning models.

**Forecasting on clean data** The most straightforward task is a simple forecast of the next $n$ timesteps, based on the last $m$ timesteps of inputs. We set $m = n = 14$. The input data into the models therefore are 14 consecutive graphs, sharing the same adjacency, or one graph with 14 node features: $(V, E, X_{i,...,i+13})$. The targets are simply $X_{i+14,...,i+27}$. As a test- and training loss we use the RMSE over all samples, nodes in $V$, and forecasted timesteps $m$.

**Stability: Noise on test data** Since usually there is heavy noise on real-world input data, it is a highly relevant and interesting scenario to test the robustness of the tested machine learning models. We adopt the experimental setting from the previous forecasting benchmarking, but add noise on the nodes features $X_{i,...,i+13}$ of the test dataset, but not onto training data. This enables us to study some aspects of the models' robustness in a controlled setting. Further, this reflects real-world data, on which noise exists but is variable due to changed measurements, or delayed reporting. We studied two different types of noise:

We found the Gaussian noise with distribution $\mathcal{N}(0, 0.01)$ to be an interesting setting for the normalized dataset. A plot of the noisy data can be found in the appendix A.2.1 Fig 9.

We also studied noise which reflects the failure of sensors/reports and therefore the occurrence of reported zeros instead of real values. Guided by this scenario, we replace $10\%$ of the test data with zeros to further study the models' robustness.

**Denoising: noise on context data** We seek to study the abilities of different models to implicitly denoise input data, by both training and testing on noisy context data. We use the same Gaussian noise and dropout noise as in the previous experiments.

We executed the benchmarking experiments each three times with different random initializations and trained all models until convergence. Details on the setup and computational aspects can be found in supplementary material A.2 or on our given GitHub. The resulting RMSEs over all samples

from the test datasets for the different tasks can be found in Table 1. A visual display of the results can also be found in Fig 10, Fig 11 and Fig 12.

Table 1: **Overview of Performance compared to different tasks.** We take the RMSEs over all samples and all forecasted time-steps from the test dataset. The $\pm$ indicates the standard deviation out of 3 training runs with random initialization respectively. We scaled all values up by the factor 100 for readability.

| Model | Forecasting | Stability | | Denoising | |
|---|---|---|---|---|---|
| | | Gauss | Dropout | Gauss | Dropout |
| Repetition | 2.27 | 2.74 | 3.30 | 2.74 | 3.30 |
| MP-PDE | $1.09 \pm 0.15$ | $2.2 \pm 0.2$ | $2.15 \pm 0.1$ | $1.37 \pm 0.05$ | $\mathbf{1.34 \pm 0.18}$ |
| RNN-GNN-Fusion | $\mathbf{0.97 \pm 0.10}$ | $\mathbf{1.3 \pm 0.1}$ | $\mathbf{1.80 \pm 0.1}$ | $\mathbf{1.33 \pm 0.06}$ | $1.48 \pm 0.11$ |
| RNN | $1.67 \pm 0.47$ | $2.3 \pm 0.3$ | $2.81 \pm 0.2$ | $1.76 \pm 0.08$ | $2.98 \pm 0.20$ |
| GraphEncoding | $3.51 \pm 0.67$ | $3.5 \pm 0.7$ | $4.10 \pm 0.5$ | $3.81 \pm 0.66$ | $5.36 \pm 1.12$ |
| TST | $1.14 \pm 0.03$ | $2.6 \pm 0.1$ | $2.5 \pm 0.02$ | $2.98 \pm 0.68$ | $2.72 \pm 0.02$ |

The RNN-GNN Fusion model, as well as the MP-PDE, generally exhibits strong performance, effectively leveraging the spatial aspect of the data. However, the performance of GraphEncoding underscores that simply incorporating a GNN does not necessarily boost performance. Therefore, meticulous benchmarking of different architectures on public datasets like ours is critical for the advancement of spatio-temporal graph machine learning.

### 4.3 TRANSFER LEARNING TO REAL-WORLD DATA

While the benchmarking of different model architectures already provides valuable insights, we aim to further illustrate the utility of synthetic data in developing foundational forecasting models for epidemiology or pre-training models on tasks on which data is sparse. We, therefore, seek to demonstrate whether pre-training on our provided synthetic data can lead to improved performance on real-world tasks. To carry out this experiment we train and evaluate the models from the previous section on real-world epidemiological data from respectively German COVID-19, German Influenza, and Brazilian COVID-19. The prediction task is again a 14-day forecast based on 14 days of input. Simultaneously, we pre-train the same models first on the synthetic dataset based on the SI-diffusion equation, and only then shortly retrain (fine-tune) them on the respective real-world data. We then compare the difference in performance. Note, that the Brazilian data requires an additional knowledge transfer onto an unseen domain (i.e. Graph). Details on the real-world datasets can be found in section A.1.4.

The outcomes of the transfer-learning experiments were highly favorable for pre-training on our dataset, underscoring the effectiveness and possible impact of our proposed method of data synthetization on real-world data and problems. The results can be found in Table 2. A graphic representation, containing the actual RMSEs and comparing the models against each other can be found in the appendix, see fig. 13.

Table 2: Change in validation-loss of the models in the transfer-learning tasks in percent. A negative number indicates improved performance through out method, i.e. by pre-training on synthetic data as opposed to training solely on the actual data.

| Model | German COVID-19 | German Influenza | Brazilian COVID-19 |
|---|---|---|---|
| MP-PDE | $-7.92\%$ | $6.96\%$ | $-16.88\%$ |
| RNN-GNN-Fusion | $-30.57\%$ | $-19.04\%$ | $-45.58\%$ |
| RNN | $-11.34\%$ | $-16.13\%$ | $-42.61\%$ |
| GraphEncoding | $-8.39\%$ | $-6.91\%$ | $-35.45\%$ |
| TST | $-11.73\%$ | $4.48\%$ | $-12.49\%$ |
| mean | $-13.99\%$ | $-6.13\%$ | $-30.60\%$ |

Only two out of 15 experiments exhibited a slightly decreased performance, while 13 showed an increased performance, frequently to a significant extent. Especially on the Brazilian COVID-19 dataset, some models showed an increase in performance up to $45\%$. Especially the previously already favorable RNN-GNN-Fusion model exhibits a significant boost in performance across all datasets.

## 5 CONCLUSIONS

We demonstrated how time-dependent PDEs can be used to create synthetic data for machine learning on graphs. In particular, we have created and published three exemplary datasets that can be used by the community for further research questions. Notably, the numerical solution of any epidemiological PDE constitutes a novelty. The described method enables researchers to create synthetic temporal graph datasets for individual use cases to support the development of new machine learning methods, in particular, for use cases where data is scarce. Even though the methodology can be easily adapted to many other use cases by using our published code, the more complex the PDEs become, the more difficult it will be to solve them, while on the other hand, simpler PDEs may lack the flexibility to model complex scenarios. However, we demonstrate through transfer-learning experiments that pre-training on our synthetic datasets can lead to improvements in real-world data performance. Despite the underlying PDE being relatively limited in flexibility and parameters, these experiments showed significant performance increases for some models. Additionally, as an application of our datasets, we have demonstrated a benchmarking of various models, showcasing that rigorous testing of common architectures was a substantial research gap due to a lack of sufficient data.

In summary, this paper presents a method for generating individual PDE-based graph datasets, offers three readily available synthetic datasets, provides a benchmarking, and highlights significance through drastic improvements of performance in real-world data. This work lays the foundation for the development of further datasets and research problems, inviting others to build upon our findings.

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

## A  APPENDIX

### A.1  DETAILS ON: DATASET CREATION

The datasets were created with different parameters of the dynamics of the PDEs. We described the key steps to numerically solve the presented equations in the main paper. Further details on the selected parameters to extend section 3 might be valuable to understand the published code and data

or to develop more suitable models and will be presented here. The following computations are executed on two AMD EPYC 7543 32-Core Processors.

The evaluation points to create the Graph $\mathcal{G}$ were created by finding the centers of the German *NUTS3* regions. Geographical data regarding the regions are publicly available. [2] The data was processed with the use of *GeoPandas* (Jordahl et al., 2020), *shapely* and *SciPy* (Virtanen et al., 2020) by finding centers of each region, adjacent regions and their distances. Some code regarding this can be found in a notebook on our GitHub under /additional_resources/point_and_mesh_generation.ipynb. The underlying graph of the wave-equation dataset was created with the same geometry files. The evaluation points were chosen randomly, the adjacencies were created using a Delaunay triangulation.

### A.1.1  SI-DIFFUSION EQUATION

The epidemiological dataset from the SI diffusion equation (1) was created by running 25 simulations, each with different parameters, with a length of $364$ time-steps. The parameters of the individual simulations can be found in Table 3.

Table 3: **Experiment Ids and their respective parameters.** The parameters $r$ and $D$ of the published 25 simulations. "Id." denotes the experiment identifier. The omitted experiment identifier corresponds to different initial values or boundary conditions and can be found in the published code for further simulations. $D$ was scaled in the table by the factor $10^{-8}$ for simplicity of the notation.

| Id. | $r$ | $D \cdot 10^{-8}$ |
|---|---|---|
| 0 | 0.6 | 2 |
| 5 | $1 + (0.2sin(5\mathrm{e}{-}6\pi x_1)sin(5\mathrm{e}{-}6\pi x_2)$ | 2 |
| 10 | $0.7 + (0.6sin(5\mathrm{e}{-}6\pi x_1)sin(5\mathrm{e}{-}6\pi x_2)$ | 2 |
| 15 | $0.5 + (0.3sin(2\mathrm{e}{-}6\pi x_1)sin(5\mathrm{e}{-}6\pi x_2)$ | 2 |
| 20 | $1.1 + (0.1sin(8\mathrm{e}{-}6\pi x_1)sin(5\mathrm{e}{-}6\pi x_2)$ | 2 |
| 25 - 45 | same as 0 - 20 | $(1 + 0.1\ sin(5\mathrm{e}{-}6\pi x_1)sin(5\mathrm{e}{-}6\pi x_2))$ |
| 50 - 70 | same as 0 - 20 | 3 |
| 75 - 95 | same as 0 - 20 | 0.8 |
| 100 - 120 | same as 0 - 20 | $(1 + 0.2sin(5\mathrm{e}{-}6\pi x_1)sin(5\mathrm{e}{-}6\pi x_2))$ |

The resulting simulations were concatenated to a single large dataset. The concatenation is possible since Infected $I$ in the individual converges to zero at the end of every individual simulation. Although the concatenation is not smooth we consider the discontinuity as negligible. A plot of the 25 different scenarios over time can be found in Figure 5.

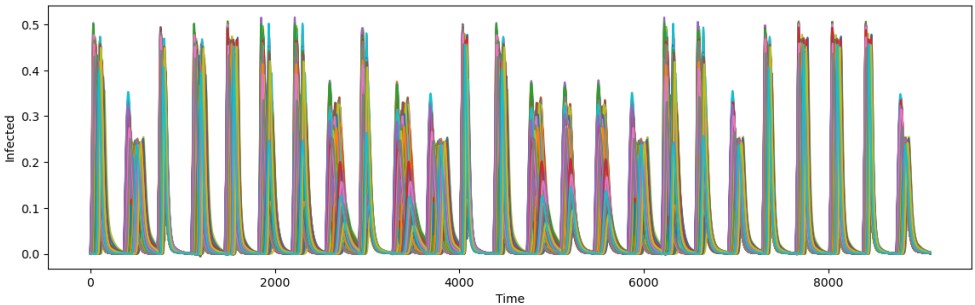

Figure 5: The dataset generated from the SI-diffusion equation over time. There are 400 lines plotted, each representing the values of an individual node over time.

---

[2]mis.bkg.bund.de/trefferanzeige?docuuid=D38F5B40-9209-4DC0-82BC-57FB575D29D7, which is published under the dl-de/by-2-0 license (https://www.govdata.de/dl-de/by-2-0 )

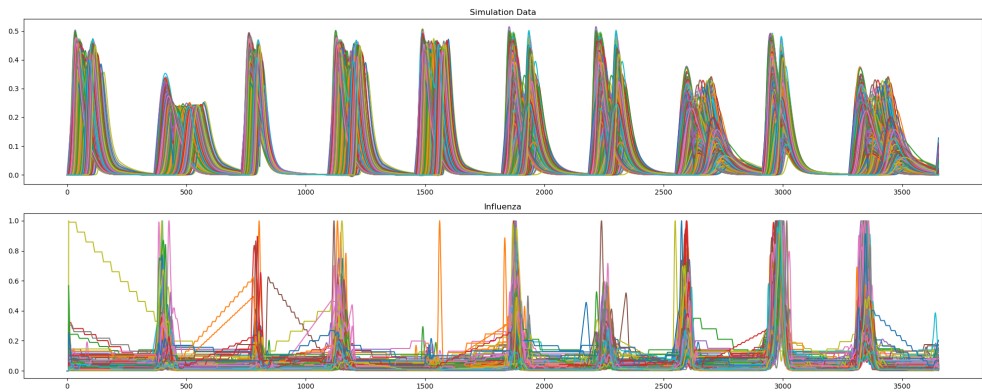

Figure 6: The SI-diffusion dataset(top) compared against the real-world German Influenza dataset(bottom). The step patterns in the Influenza dataset are presumably artefacts of the data collection/management pipeline.

### A.1.2 ADVECTION-DIFFUSION EQUATION

The dataset created from the advection-diffusion equation (2) consists of a single, connected, but long simulation. We define $54$ different but consecutive sets of dynamics of the equation, each with a length of $80$ time-steps. We set $h = \frac{10^9}{8}$, $T = 54 \cdot 10^{10}$.

The support, necessary to describe the source term $s(x,t)$, $\Omega_{s(x,t)}$, is a rectangle $[a \cdot 10^6, 1.01a \cdot 10^6] \times [b \cdot 10^6, 1.01b \cdot 10^6]$ on the domain.

The source term $s(x,t)$ was specified in the main paper via $S(t)$ which can be recovered by scaling the temporal dimension $S(t) = \hat{s}(t \cdot 10^{-10}) \cdot 10^{-10}$ from Table 4 as scaled characteristic function, i.e.

$$\mathbb{1}_I(t) = \begin{cases} 1 & t \in I \\ 0 & t \notin I. \end{cases},$$

Where $I \in \mathbb{R}$ is some interval. Also $\beta(x,t)$, and $a, b$ can be found in Table 4.

Table 4: **Parameters of the advection-diffusion equation** The parameters of the simulation for the advection-diffusion equation."Id." represents 80 consecutive time-steps. The factor $10^4$ in the column defined $\beta$ was used for simplicity of the notation. Note that in the characteristic functions $\mathbb{1}$ also a modulus operation is used, such that after Experiment Id 6, the same dynamics as in 1 starts, i.e. the intervals were shifted to the right. TODO Jost Ids. 7-12 does not match first columns

| Id. | $\hat{s}(t)$ | $\Omega_s$ | $\beta(x) \cdot 10^4$ |
|---|---|---|---|
| 1 | $-32\mathbb{1}_{[0,0.1]} - 22\mathbb{1}_{[0.8,0.9]}$ | $a = 3.45, b = 5.4$ | $(-0.5, 0.5)$ |
| 2 | $-27\mathbb{1}_{[1.3,1.4]}$ | same as 1 | same as 1 |
| 3 | $-32\mathbb{1}_{[2.1,2.2]} - 42\mathbb{1}_{[2.8,2.9]}$ | same as 1 | same as 1 |
| 4 | $-28\mathbb{1}_{[3.3,3.4]}$ | same as 1 | same as 1 |
| 5 | $-38\mathbb{1}_{[4.1,4.2]} - 33\mathbb{1}_{[4.8,4.9]}$ | same as 1 | same as 1 |
| 6 | $-30\mathbb{1}_{[5.3,5.4]}$ | same as 1 | same as 1 |
| 7-12 | same as 1-3 | $a = 3.6, b = 5.6$ | same as 1 |
| 13-18 | same as 1-3 | $a = 3.45, b = 5.8$ | same as 1 |
| 19-36 | same as 1-18 | same as 1-18 | $(0.6, 55 - x_1 10^{-6})$ |
| 37-54 | same as 1-18 | same as 1-18 | $(0.5, 2.5x_1 10^{-6} - 1.375)$ |

Similar to the source term $s$, we also initialize once $u$ on the support $\Omega_{s(x,0)}$ with

$$u = \begin{cases} 1 & x \in \Omega_{s(x,0)} \\ 0 & x \notin \Omega_{s(x,0)}. \end{cases}$$

A spatio-temporal visualization of the advection-diffusion equation can be found in Figure 7

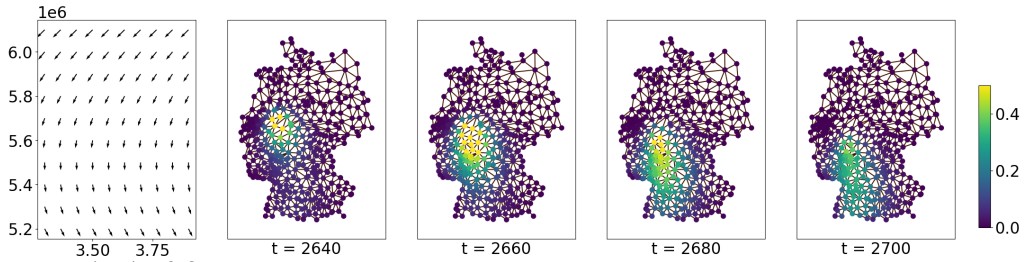

Figure 7: Plot of some advection-field on the left side and an exemplary plot of the solution of the advection-diffusion equation over 4 time-steps.

### A.1.3 Wave equation

The wave equation (3) also consists of a single consecutive simulation. There are two waves throughout the simulation, that start at the boundaries. The boundary conditions are usuallyRobin boundary conditions $\alpha u + \beta \eta \cdot \nabla u = 0$ three exceptions: first during the interval $0 < t < 50000$and on the boundary with $x_1 > 6200000, x_0 < 3400000$, second during the interval $1000000 < t < 1100000$, and on the boundary with $x_1 > 6200000, x_0 > 3800000$, and third during the interval $1800000 < t < 1900000$, and on the boundary with $x_1 > 6200000, 3500000 < x_0 < 3600000$.

On these intervals and regions, the Dirichlet boundary $g$ are respectively a constant factor $(0.8, 1, 0.9)$ for the wave's amplitude, that can easily be adapted for further simulations. we set $b = 0.000005$. The first wave can be seen in Figure 4.

### A.1.4 Creation of epidemiological real-world datasets

The three real-world datasets from section 3 are based from epidemiological data, which can be found online.

1. The Brazilian COVID-19 dataset has 27 nodes and 1093 time-steps spanning 2019-2022 after concatenation and linear interpolation for daily resolution and is publicly accessible. [3]

2. German COVID-19 data can be found at github.com/robert-koch-institut/SARS-CoV-2-Infektionen_in_Deutschland, with 1539 time-steps.

3. German Influenza dataset can be found at survstat.rki.de/ with our curation having 5256 time-steps.

All 3 datasets were smoothed with a 7-day moving average.

The graph describing the spatial connection of the *NUTS3* regions in Germany was created above. The Brazilian geospatial data used for constructing the graph connectivity as described in section A.1 can be accessed at https://www.ibge.gov.br/en/geosciences/territorial-organization/territorial-meshes/18890-municipal-mesh.html.

A visualization can be found in Fig. 8.

---

[3]sisaps.saude.gov.br/painelsaps/atendimento

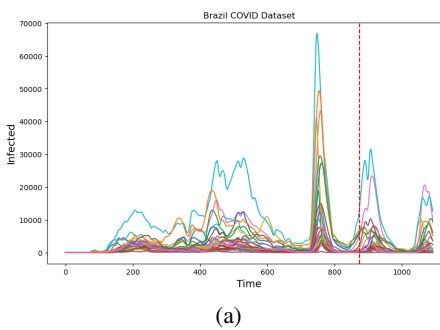 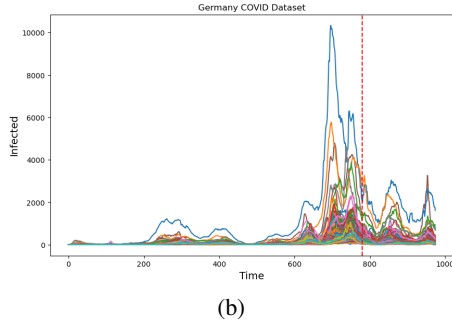

(a)                                          (b)

Figure 8: Left: Brazilian COVID dataset. Right: German COVID dataset. In both plots, each curve represents a region, and the vertical red line indicates the training/evaluation cutoff.

These datasets will additionally be released upon acceptance.

## A.2 DETAILS ON: APPLICATIONS OF SYNTHETIC DATA

In our experiments for epidemiological applications from section 4, we trained and evaluated the models on the data based on the SI-diffusion equation in 32-bit float representations. The employed training epochs for the different tasks and models can be found in Table 5.

Table 5: **Overview of Training Epochs and Computational Aspects for Different Models.** Number of parameters, their respective training times for the forecasting benchmarking, and maximum training epochs for all experiments from section 4 for each model. The exact training times and epochs depend on the specific training run due to random initialization and early stopping.

| Model | # Params | Train Time | #Epochs |
|---|---|---|---|
| MP-PDE | 623k | 57m | 30 |
| RNN-GNN-Fusion | 70k | 2:29h | 30 |
| RNN | 67k | 1:16h | 30 |
| GraphEncoding | 382k | 3:42h | 30 |
| TST | 344k | 1:51h | 50 |

The exact hyperparameters can be found in the implementation on GitHub in the regarding parameter files such as github.com/github-usr-ano/Temporal_Graph_Data_PDEs/blob/main/ml/mp_pde/mp_pde.yml for the MP-PDE model. The parameter files for other models can be found in their respective directories. The hyperparameters were chosen as optimal for each model experimentally. We trained all models for each task with the Adam optimizer until convergence (early stopping).

### A.2.1 DETAILS ON EPIDEMIOLOGICAL BENCHMARKING

A visualization of some data with Gaussian noise from the stability and denoising experiments can be seen in Figure 9.

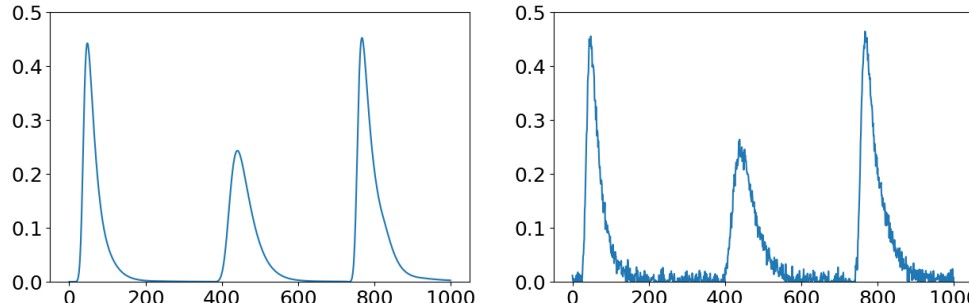

Figure 9: Illustration of the effect of additive noise sampled from the distribution $\mathcal{N}(0, 0.01)$ from task 2 and task 4. On the left is the time series from a random graph node; on the right is its respective noise-added time series. The plotted section is the first 1000 time-steps of the training dataset.

A visual comparison of RMSEs for each time-step in the forecasting experiment for different models can be seen in Fig 10

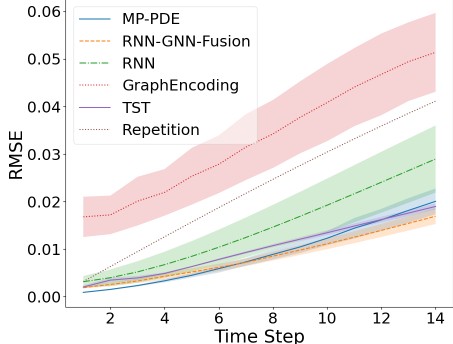

Figure 10: The RMSE over all samples from the test dataset. The less saturated colors represent the standard deviation for different initializations.

A visual comparison of RMSEs for each time-step in the stability experiments for different models can be seen in Fig 11.

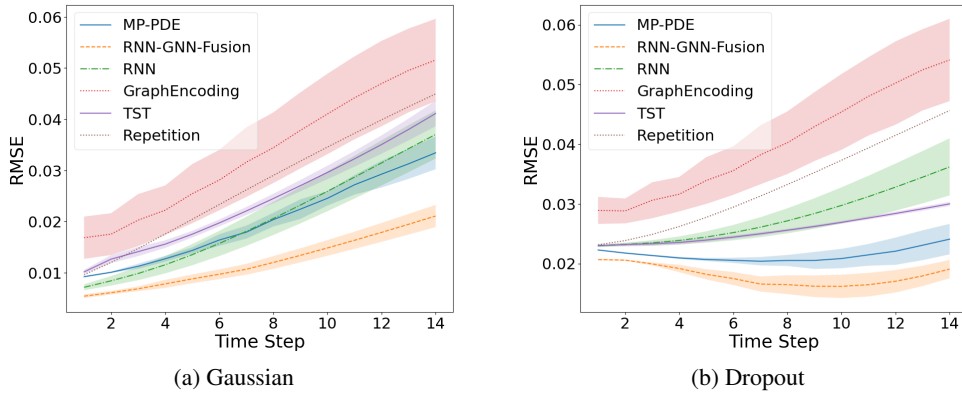

Figure 11: Plots of the RMSE of the presented models during the stability experiments for each predicted time-step. The models were trained on the synthetic dataset, and evaluated on a synthetic dataset with: **a)** additive Gaussian noise, **b)** dropout noise. The less saturated colors represent the standard deviation for different initializations.

A visual comparison of RMSEs for each time-step in the denoising experiments for different models can be seen in Fig 12.

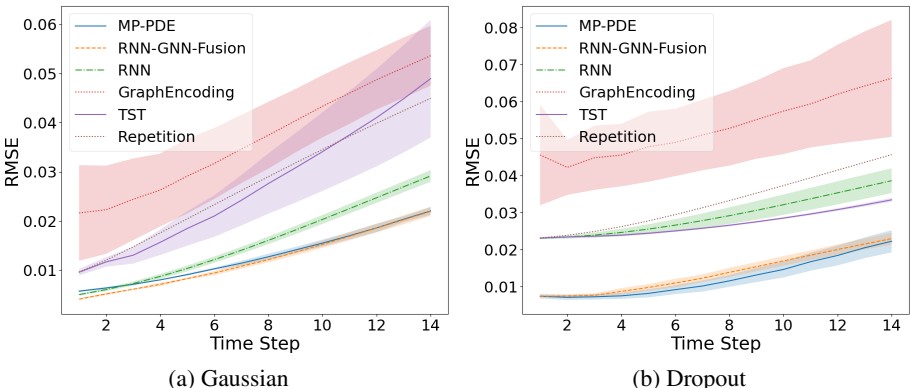

(a) Gaussian          (b) Dropout

Figure 12: Plots of the RMSE of the presented models during the denoising experiments for each predicted time-step. The models were trained and evaluated on a synthetic dataset which. The models were trained to predict based on context data, with **a)** additive Gaussian noise, **b)** dropout noise. The less saturated colors represent the standard deviation for different initializations.

### A.2.2 DETAILS ON TRANSFER LEARNING TO REAL-WORLD DATA

To have a better visual comparison between the predictive performances of the models and the degree of their improvement in the transfer learning experiment, we created Fig. 13.

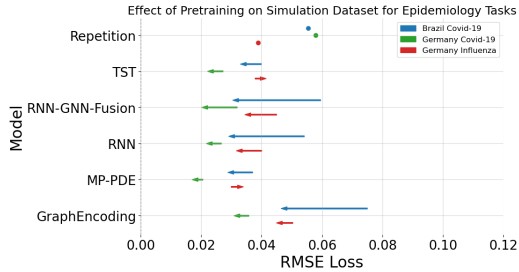

Figure 13: Plots of the RMSEs of the presented models during the transfer-learning experiments. The arrows point from non-pre-trained to pre-trained models.

### A.2.3 ADDITIONAL EXPERIMENTS

We repeated the Forecasting experiment from section 4, more specifically section 4.2 on the two other created datasets, i.e. the Advection-diffusion dataset and the wave-equation dataset.

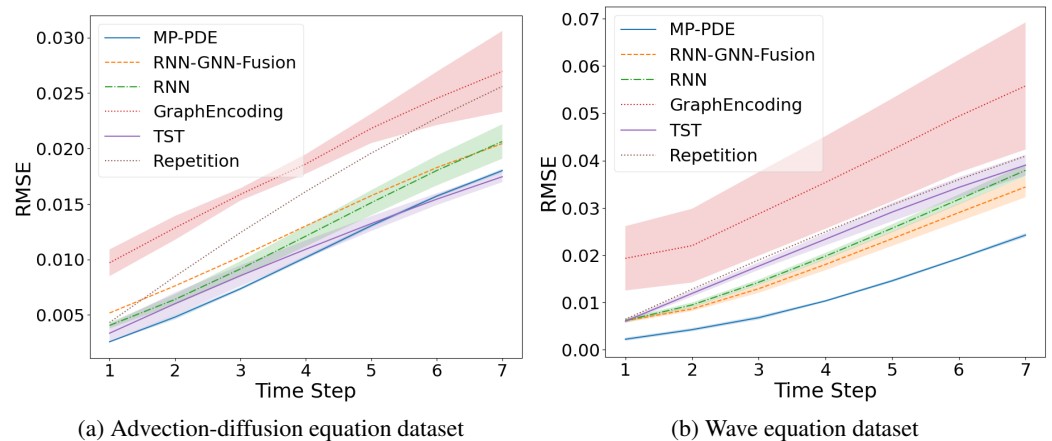

(a) Advection-diffusion equation dataset      (b) Wave equation dataset

Figure 14: **a)** The RMSE for 7-timestep forecasts on the dataset generated by Advection-Diffusion Equation. **b)** The RMSE for 7-timestep forecasts on the dataset generated by Wave Equation.

Table 6: **Performance Comparison Across Models for Different Datasets**

| Model | Advection-Diffusion Equation | Wave Equation |
|---|---|---|
| Repetition | $0.0156 \pm 0$ | $0.0244 \pm 0$ |
| MP-PDE | $\mathbf{0.0115 \pm 0.0001}$ | $\mathbf{0.0139 \pm 0.0003}$ |
| RNN-GNN-Fusion | $0.0134 \pm 0.0009$ | $0.0213 \pm 0.0013$ |
| RNN | $0.0135 \pm 0.0010$ | $0.0234 \pm 0.0008$ |
| GraphEncoding | $0.0195 \pm 0.0012$ | $0.0383 \pm 0.0101$ |
| TST | $0.0117 \pm 0.0006$ | $0.0256 \pm 0.0014$ |

