# OpenReview forum: "Synthetic Datasets for Machine Learning on Spatio-Temporal Graphs using PDEs"
_ICLR.cc/2025/Conference — Submitted to ICLR 2025_

### Official Review · Reviewer_Jv2x · 2024-10-28

**Soundness:** 3
**Presentation:** 3
**Contribution:** 3
**Rating:** 6
**Confidence:** 4

**Summary:**

This paper presents a methodology for generating synthetic spatio-temporal graph datasets using partial differential equations (PDEs). The authors illustrate their approach by creating three datasets that model disaster scenarios: the spread of epidemics, atmospheric particle movement, and tsunami propagation. The main contributions of this work include (1) a detailed methodology that employs the Finite Element Method to convert PDE solutions into graph data, (2) three ready-to-use synthetic datasets, (3) an open-source implementation for creating custom datasets, and (4) empirical validation through benchmarking and transfer learning experiments.

**Strengths:**

- The PDE solving methodology demonstrates mathematical rigor through detailed documentation and established numerical methods.
- The authors provide practical resources through ready-to-use datasets and accompanying code for the research community.
- The experimental results strongly validate the approach by showing significant benefits in transfer learning tasks.
- The framework enables researchers to create custom datasets by modifying parameters and PDEs for different applications.

**Weaknesses:**

- The paper only demonstrates relatively simple PDEs despite covering three different scenarios.
- The transfer learning experiments are limited to epidemiological applications without exploring other real-world domains.
- The authors do not thoroughly analyze how variations in PDE parameters impact the learning outcomes.

**Questions:**

1. Have you explored how the choice of PDE parameters affects the quality of the synthetic data for transfer learning?
2. Could this approach be extended to more complex PDEs with coupling or higher-order terms?
3. How do the computational requirements scale with domain size and mesh resolution?
4. Have you considered applications beyond the disaster scenarios presented?

---

> ### Author Response · Authors · 2024-11-18
>
> We would like to thank the reviewer for this thorough review and feedback, and appreciate the already provided rating. We would like to address the weaknesses and questions raised by the reviewer:
>
> **Weaknesses:**
> 1. Our transfer learning experiment show drastically increased performances on real-world tasks (up to 45% Table 2).  Thus, we see no necessity to artificially increase the PDEs complexity. Additional complexity, would lower the readability of our paper and code with limited chances of increasing its general utility.
>    But, by using the FEM (the probably most versatile and advanced numerical method for PDEs), our method works evenly fine for very complex PDEs such as the Navier-Stokes, Cahn-Hillard, or Schrödinger equation. Our sections 2 and 3 enable readers to arbitrarily adapt our method to implement individual complexity - we will elaborate on the next paragraph.
>
> 2. To conduct transfer-learning experiments from these datasets, suitable real-world datasets do not exist, as, for example, simply not enough tsunami-waves were measured and sampled.
>    This paper was initiated upon realising that epidemiological spatio-temporal data is insufficient. Thus we created the published epidemiological dataset. To further demonstrate our method, we chose the other PDEs not only  due to their focus on disaster scenarios, but also to include additional prominent PDE-terms, that were not already covered in the SI-Diffusion equation: 1) a spatial derivative of order one (gradient), 2) a time-derivative of order two, 3) linear PDEs, and 4) Robin and Neumann boundary  conditions. Thus, both the Advection-Diffusion equation and the Wave equation suited perfectly to elaborate on our method. The resulting two datasets aim to elaborate on our method and support theoretical-, explorational- and preliminary studies and benchmarking.
>
> 3. Our paper proposes a method for data-generation, creates three datasets for demonstration, executes exemplary benchmarking on them and shows its utility through transfer learning to real-world data with strong results.
>    Thus, we offer a tool to create datasets and datasets itself, similar to PDEBench. Papers that aim to study the models behaviour (like PINNs) under variation of parameters, can build upon our work. To us, further research towards a deeper understanding of the interaction of PDEs parameters and models performances seems interesting, but is out of the scope of our paper.
>
> **Questions**
> 1. The epidemiological parameters were chosen to be meaningful from an epidemiological perspective, not to gloss over the presented results. There was no cherry picking in parameters, diseases or countries. This highlights the potential of our method for upcoming diseases and other applications, since without exact knowledge of the parameters great results were achieved.
> 2. Yes. Our method can be applied to all PDEs for which the FEM can be used. The FEM is known to be the most flexible and advanced numerical technique for all kind of PDEs.
> 3. Domain size does not influence computational requirements. Note, that there are two different meshes involved: i) a mesh used to solve a PDE, and ii) a set of points to create a graph. (See Fig 1.)
>    We are not sure which mesh the reviewer refers to, so we answer for both meshes:
>    i) The pure FEM generally does not scale well for high mesh resolutions, but there are many methods solving this issue like sparse matrices, iterative matrix solver and adaptive mesh refinement.
>    ii) The creation of much larger graphs does not heavily influence the computational complexity (currently it scales linear with a good scaling factor, but could even be parallelised). In fact we give a brief explanation in our GitHub under *Changing the sampling points on a domain* on how to increase the graph size, and provide an additional dataset with a much larger graph.
> 4. Yes, but see answer to weakness 3.
>
> We would like to thank the reviewer for the very thoughtful remarks and questions. Coming from an epidemiological perspective, we know that there is a significant need and interest within the community for our dataset. We believe we have addressed the reviewer's objections and have further elaborated on the benefits of our work through the responses provided. If the reviewer is satisfied with our responses, we kindly ask the reviewer to consider raising their score to enhance our chances of acceptance and make our datasets and methods accessible to other researchers.

---

> > ### Author Response · Authors · 2024-11-22
> >
> > Dear Reviewer Jv2x,
> >
> > As we are approaching the end of the discussion period we would kindly remind you to have a look at our provided answer. Please do not hesitate to let us know, if there are any questions or if there is still anything unclear. We greatly look forward to your important response.
> >
> > Best regards,
> >
> > The authors

---

### Official Review · Reviewer_vQRP · 2024-10-31

**Soundness:** 1
**Presentation:** 1
**Contribution:** 1
**Rating:** 3
**Confidence:** 5

**Summary:**

In this work, the authors generate synthetic spatio-temporal graphs with fixed structure and evolving node features by solving PDEs.

**Strengths:**

- The authors identified the gap in the literature, wherein high quality temporal graph datasets are not abundant.

**Weaknesses:**

### `(W1) Benchmarking`
The authors use the following models:
- Repetition (naive)
- RNN (classic)
- TST (Transformer)
- MP-PDE (modified baseline)
- RNN-GNN-Fusion (source of this model is not clear)
- GraphEncoding (modified baseline)

They report the performance of these models on the synthetic datasets in Table 1 (Forecasting column), where the model `GraphEncoding` performs worse than the naive baseline `Repetition`. This is an odd observation, and better baselines should've been used for benchmarking.

It is recommended to check the paper **Graph-based Multi-ODE Neural Networks for Spatio-Temporal Traffic Forecasting** (TMLR) and the baselines used there. For example, the authors could test out the benchmarks `STGODE`, `GRAMODE`, and `ARIMA` on the generated datasets.

### `(W2) Contribution Claim`
The PDEs used to create the datasets are not novel, nor is the technique to solve them. The authors claim:
> This or any similar epidemiological PDE, based on the SIR-ODE (Kermack & McKendrick, 1991), has never been solved numerically

And then in footnote 1 on page 4, the authors mention:
> The methodology we use here as well as the numerical code can be found in this tutorial of the used library https://www.dealii.org/current/doxygen/deal.II/step_23.html. We made only smaller adaptions to the code.

It is not clear what is the contribution of this work, apart from setting hyper-parameters of known PDEs and solving them numerically using known libraries.

### `(W3) Lack of new insights`
Consider this experiment: there are n number of models $M_1, M_2, \cdots, M_n$ and a standard dataset $D$ which is used in the literature. When the models are tested on this dataset, it results in a ranking $r_1, r_2, \cdots, r_n$. Now consider a synthetic dataset $D'$ which when used for benchmarking the models results in the ranking $r_1', r_2', \cdots, r_n'$. Comparing these two rankings can highlight the use of the synthetic dataset $D'$ and the authors could comment on the reason for such change in ranking, if any. On the other hand, if there is no change in the ranking, then the dataset $D'$ brings no new insights.

It is recommended that the authors run this experiment using at least two standard spatio-temporal traffic datasets `METRLA` and `PEMSBAY`.

### `(W4) Organization of the contents`
The content in the paper is not well organized. The focus was moving from one topic to another without a smooth logical transition. Here are some thoughts on an alternative structure for the paper:
- In **Introduction** the authors only discuss the problem they are trying to solve and dedicate a clear paragraph to the motivation, then in a subsection they discuss the **related works**, as in what has been tried before in the same setting.
- Then, in a **Methodology** section, the authors should expand on their proposed technique using a block diagram giving a higher-level overview of the work, and present a detailed version through a step-by-step algorithm. The PDE should be kept generic, and the three special cases can just be mentioned after the algorithm is presented
- Then, a section on **Experiments** should be included to specify clearly what the authors what to conduct, and divide experiments into self-contained subsections, with highlighted key insights derived from the results. Here, the authors may mention the baselines being used and delegate further information on their implementation/modification to the Appendix for those interested.
- The figures with **dataset examples** should be deferred to the Appendix as well.
- In the conclusion section, the authors should clearly state the **limitations** of the work, and what could be potentially improved in the future.
- Some results which are currently in the Appendix should be brought to the main body.

### `(W5) Limitations of the generated datasets`
The graph structure is fixed, and the dimensionality of the node feature is limited to 2.

**Questions:**

### `(Q1) Transfer Learning` (Sec. 4.3)
- Could the authors please elaborate on the meaning of fine tuning?
- How does the chosen (modified) baselines used in this paper compare against the standard spatio-temporal benchmarks, for example `GraphWaveNet`, and `STGODE`?
- Why did the authors only report validation loss?
- What is the test RMSE on (1) training a model $M$ on real data $D$, and (2) training the model $M$ on real data $D$ after pre-training on synthetic data $D'$?
- How do the models compare against the naive repetition baseline?

### `(Q2) Noise` (Sec. 4.2)
The authors mention:
> We found the Gaussian noise with distribution $\mathcal{N}(0, 0.01)$ to be an interesting setting for the normalized dataset.

Could they please provide their rationale for choosing the specific noise variance? The explanation would aid in understanding the experiment better.

- The figures 10-12 are not clear; Why is there a drop in RMSE for dropout noise? Are the zero values emulating missing sensor data considered or ignored during evaluation? In the literature, zeros are treated as missing values and ignored during training and evaluation.
- In Table 1, it is better to report the relative change in RMSE. That would show that the naive repetition baseline is robust to the Gaussian noise considered in the study.

----

### `Feedback for improvement`
- The paper needs to be reorganized and re-written for clarity
- Relevant baselines for node feature forecasting should be used for benchmarking
- The benchmarking results should be contrasted with the ranking on standard datasets to deliver new insights
- The transfer learning and noise robustness experiments need to be clarified and more details should be added
- The experiment setup should be explained more clearly
- The limitations of the work should be admitted
- The contributions should be stated clearly
- A scalability section can be added, where synthetic datasets can be generated for increasing number of nodes, and the relative performance of the baseline models can be reported across the varying size.

---

> ### Author Response · Authors · 2024-11-20
>
> We thank the reviewer for his very detailed feedback and want to address the weaknesses and questions
>
> ## W1
> The benchmarking in section 4.2 is an exemplary application of our dataset. We do not seek to promote a certain model, but aimed to compare performance aspects of different architectures, while also having a naive baseline as a comparison. We provide a flexible tool and with it some exemplary datasets that can be used by other researcher to test and imrpove their models for scientific purposes or as we have shown in Section 4.3 for real word applications.
>
> The initially tested models are all capable to be transfered onto other graphs, while the models suggested by the reviewer are all bound to a fixed structure. Also they are specialised for traffic-predictions, not optimised for epidemiological tasks. We took the implementations from the respective GitHubs and conducted the suggested experiments, but no hyperparameter-tuning or other optimisation could be done in time. We prefer not to include them in our paper, as, by our experiments, cf. Table below, the performance of these models do not transfer well from the initial results on traffic data to epidemiological data and it is not our purpose to advocate a particular method.
>
> The results of these experiments and their computational aspects can be found in the second comment below.
> The poor performances underline the importance of specialised datasets like ours and careful benchmarking.
>
> ## W2
> Footnote 1 concerns only parts of section 2.3. The novelty regarding the epidemiological PDE (section 2.1) does not conflict the fact, that the wave equation has been solved before. Also the construction of the datasets, through the creation of a graph, adjacencies and distances is a novelty. Currently no PDE-based temporal graph dataset exists. To help the reviewer understand the impact and difficulty of our contributions, we refer to PDEBench, which has successfully done something similar.
> ## W3
> This only concerns section 4.2 and the appendix. Since indeed the benchmarking results on the other two datasets show minimal differences, we included them in the appendix.
> No prior standard dataset and therefore neither rigorous benchmarking, nor a similar transfer-learning experiment exists.
>
> Comparisons to *PEMS-BAY* or *METR-LA* are unsuitable because they are neither synthetic, nor PDE-based, nor related to our applications. Benchmarking the used models on entirely different datasets is beyond the scope of our paper.
>
> ## W4
> We appreciate the suggestions, but note that ICLR doesn't mandate the **IMRaD** format. A change in the structure of our paper would be a significant change and does not seem to be necessary, as the other reviewers all rated our presentation with "3: Good".
>
> ## W5
> It is unclear to us, what "fixed graph structure" refers to:
> We create two different graphs (3rd on GitHub), and evaluate the models on yet another graph. Neither our method, nor the datasets, nor the ML models fix the graph structure.
> Different graphs for the same PDE would create more redundancy in the datasets (see W3), while excluding some ML models (e.g. the ones suggested by the reviewer require fixed graphs). Moreover, the epidemiological graph structure represents the real-world structure, which is also fixed.
>
> Indeed, the dimensionality is limited to 2, but we see no necessity to artificially increase its complexity as the Table 2 displays already an enormous boost in real-world performance. Additional complexity would lower the readability of our paper and code with limited chances of increasing its general utility. Our method enables others to arbitrarily increase $d$.
>
> ## Q1
> - Fine-tuning refers to training an ML model on real world data, which has been trained on synthetic data before. See en.wikipedia.org/wiki/Fine-tuning_(deep_learning)
> - See W1
> - Since some models used teacher-forcing while others were trained auto-regressively, and all employing different dropout, training losses do not benefit the reader
> - This is the transfer learning experiment from section 4.3. RMSEs can be found in Fig. 13.
> - We updated Fig. 13.
> ## Q2
> The noise was chosen by a visual comparison of Fig6, 8 and 9
>
> - Some models propagate information to nodes, that were set to zero after a few time-steps. The zero values were not ignored, as in an epidemiological real-world scenario i): it is not possible to know whether the zero is a false reporting or actual measurements, ii)  missing data often contains a value close to zero no actual zero.
> - Relative changes skew the presentation towards poorly performing models.
>
> We want to thank the reviewer for the thoroughly review and feedback. We are confident, that our rebuttal could clarify many points and have included all models for benchmarking requested by the reviewer. Therefore we kindly ask the reviewer to reconsider their rating of our paper to grant other researchers access to our datasets, methods and benchmarks.

---

> > ### Author Response · Authors · 2024-11-20
> >
> > ## Appendix to previous comment:
> >
> > Additional experiments for Weakness 1, compared to Table 1 from our paper:
> >
> > | Model          | Forecasting              | Stab(G)                | Stab(D)                 | Denois(G)                | Denois(D)                |
> > | -------------- | ------------------------ | ---------------------- | ----------------------- | ------------------------ | ------------------------ |
> > | Repetition     | $2.27$                   | $2.74$                 | $3.30$                  | $2.74$                   | $3.30$                   |
> > | MP-PDE         | $1.09 \pm 0.15$          | $2.2 \pm 0.2$          | $2.15 \pm 0.1$          | $1.37 \pm 0.05$          | $\mathbf{1.34 \pm 0.18}$ |
> > | RNN-GNN-Fusion | $\mathbf{0.97 \pm 0.10}$ | $\mathbf{1.3 \pm 0.1}$ | $\mathbf{1.80 \pm 0.1}$ | $\mathbf{1.33 \pm 0.06}$ | $1.48 \pm 0.11$          |
> > | RNN            | $1.67 \pm 0.47$          | $2.3 \pm 0.3$          | $2.81 \pm 0.2$          | $1.76 \pm 0.08$          | $2.98 \pm 0.20$          |
> > | GraphEncoding  | $3.51 \pm 0.67$          | $3.5 \pm 0.7$          | $4.10 \pm 0.5$          | $3.81 \pm 0.66$          | $5.36 \pm 1.12$          |
> > | TST            | $1.14\pm 0.03$           | $2.6\pm 0.1$           | $2.5 \pm 0.02$          | $2.98 \pm 0.68$          | $2.72 \pm 0.02$          |
> > | GraphWaveNet   | $2.72 \pm 1.05$          | $4.05 \pm 0.57$        | $4.05 \pm 0.58$         | $2.48 \pm 0.31$          | $3.71 \pm 1.05$          |
> > | STGODE         | $6.40 \pm 1.29$          | $6.40 \pm 1.23$        | $6.62 \pm 1.22$         | $6.80 \pm 2.31$          | $6.93 \pm 1.51$          |
> > | gramODE        | $2.51$                   | $2.59$                 | $2.6$                   | $2.6$                    | $48.19$                  |
> >
> > and computational aspects, compared to Table 5 from our paper:
> >
> > | Model          | \# Params | Train Time | \#Epochs |
> > |----------------|-----------|------------|----------|
> > | MP-PDE         | $623$K    | $57$m      | $30$     |
> > | RNN-GNN-Fusion | $70$K     | $2$:$29$h  | $30$     |
> > | RNN            | $67$K     | $1$:$16$h  | $30$     |
> > | GraphEncoding  | $382$K    | $3$:$42$h  | $30$     |
> > | TST            | $344$K    | $1$:$51$h  | $50$     |
> > | GraphWaveNet   | $265$K    | $2$:$02$h  | $30$     |
> > | STGODE         | $616$K    | $25$m      | $30$     |
> > | gramODE        | $9$M      | $23$:$20$h | $30$     |

---

> > ### Comment · Reviewer_vQRP · 2024-11-20
> >
> > ## `(W1) Benchmarking`
> > The baselines suggested operate through Ordinary Differential equations. They are tested on traffic datasets, but are not designed specifically for them. I understand that the authors are not proposing a model, yet benchmarking on the proposed datasets should be done properly using standard models from the literature, and not creating something like RNN-GNN-Fusion as standard RNN-GNN architectures exist already. As the authors admit that due to lack of time, the hyperparameters could not be tuned properly, the results provided in the table are inadmissible. Moreover, GRAM-ODE and STG-ODE are known to outperform GraphWaveNet in peer-reviewed papers (check the same reference given in the original comment). Therefore, `the benchmarking setup cannot be considered valid, until standard benchmarks are used.` The authors did not comment on the absurd performance of the `naive repetition baseline`. It must also be noted
> >
> > ## `(W2) Contribution Claim`
> > As I understand, and please correct me if I'm wrong, `the contribution is limited to generating a fixed graph structure (mesh) on which the PDEs are then solved`. If the authors have improved upon **PDEBench** it should be highlighted in the paper clearly to highlight some contribution.
> >
> > ## `(W3) Lack of new insights`
> > The experiment was only a suggestion, which can strengthen the validity of the proposed datasets.
> >
> > ## `(W4) Organization of the contents`
> > This was also just a suggestion; while IMRaD is not enforced by ICLR, it allows for a smooth progression and logical structure of the paper. `The rating on presentation is completely based on my perception, and the score given by other reviewers does not have any impact whasoever.`
> >
> > ## `(W5) Limitations`
> > By fixed graph structure, I mean that the adjacency matrix of the graph does not change over time. Since the work is mostly motivated through transferability of the synthetic datasets, `would the datasets with node dimension 2 help transfer to real datasets with node dimension k > 2?` If not, this is a reasonable limitation and should be admitted in the paper.
> >
> > ## `(Q1) `
> > - The term "fine tuning" is not exclusively used for transfer learning, and therefore should be clarified in the paper. Moreover, the details of fine tuning, and how the pre-training and training was done, should be mentioned.
> > - What about the test loss, as that potentially has more samples than the validation set.
> > - the details of the real world datasets is not presented clearly. For Brazil dataset, it is said that it has 27 nodes, but for other datasets nothing is mentioned in A.1.4. `Could the authors please mention the number of nodes in the real world datasets?` This could potentially explain the big jump in performance for Brazil as it has small number of nodes, and therefore easier to train.
> >
> > ## `(Q2)`
> > the noise robustness analysis is not done thoroughly, as the results should be reported for varying noise levels.

---

> ### Author Response · Authors · 2024-11-21
>
> We thank the reviewer for the timely feedback to our rebuttal. We are happy and eager to discuss, but it is important to mention again, that our paper proposes a method to generate synthetic datasets based on well studied PDEs (we have provided datasets for different examples) that can be used for highly relevant ML tasks, such as benchmarking (for which we have demonstrated the compatability for various well-known models), pre-training models for real-world applications (for which we have showed great model improvement to forecast actual infection data), and more.
> ## W1
> *RNN-GNN-Fusion* is a simple implementation of a _time-and-graph_ model as described by (Gao & Ribeiro, 2022), which they refer to as "the most common representation adopted in literature". Our implementation choses GRUs as the RNN, and MPNN as GNNs; two common choices. This is a standard model.
>
> As we used the implementations from GItHub with a hardcoded (similar to ours) amount of parameters, we do believe our benchmarking results are admissible. The results emphasise the need for datasets outside the traffic domain. The results rather invite and motivate model proponents to adapt their models for the provided datasets, or on other PDE-based datasets that are generated by this method for an application of choice.
> The reviewer called the naive repetitions models results "absurd" and "odd" - however we think this again displays the strong need to benchmark the models named by the reviewers on our datasets.
> To get an understanding of the "naive repetition" models performance, Fig 6. might help: after an infection wave, a repetition delivers good results until the next wave. This usually concerns a longer time-period. Also shortly before the top of a wave, a repetition model performs well. The only two phases of an infection waves, on which the repetition model performs very poor, are rise and fall.
> ## W2
> Contrary to the reviewers statement, the contribution is not limited to generating fixed graph structures on which the PDEs are solved. A PDE is solved using a mesh, but evaluated on a different set of points. This different set of points forms a graph and can easily change between every timestep. In fact, the adjacencies are computed in a different and later step, independent of the PDE-solver. It is even possible to create different graphs at the same time-step without recomputing the solution or the FEM-mesh. The graph structure in the datasets is fixed only to align with real-world data, not due to limitations of our method.
>
> The outcome of our methods are irregular  temporal `graphs`, equipped with different distances, on flexible domains, to match real-world scenarios. PDEBench provides grids (e.g., 512x512).
> Further PDEBench offers (2D)spatio-temporal data for the following famous PDEs
> - Diffusion-Reaction
> - (in)compressible Navier Stokes
> - Darcy Flow
> - shallow water
>
> We offer
> - SI-diffusion (novelty)
> - wave
> - advection-diffusion
>
> Further PDEBench employs different methods, mostly finite volumes, specialised for respective PDEs. We use the FEM and suggest to adapt our code and method for custom applications, geometries, domains and PDEs.
> ## W3
> /
> ## W4
> /
> ## W5
> The adjacency matrices of our graph datasets do not change over time, because the real-world data, which we tried to align, does not (counties are not moving). With few adaptions (if this use-case exists), our method enables others to quickly do this without even changing the underling PDE/solver, cf W2.
>
> If we understand the reviewer correctly, the question is whether node-dimension 2 in synthetic data is sufficient to improve real-world performance. The answer is yes, thus this is not a limitation. This is exactly what we showed in our transfer-learning experiment: the models are trained with node-dimension 1, while in reality the relevant dimension might be much higher (Infected, population-density, mean age, vaccination status, etc). Still, the performance increases drastically. We hope this proves the point.
>
> Our method does not limit the node dimension, usually real world data does. It is not difficult to extend the epidemiological PDE, but we see no need to artificially increase the feature size.
> In other applications, other PDEs might be used, which is supported by our method. Thus, this is neither a weakness of our datasets, nor our method.
> ## Q1
> - Thanks, we have slightly changed our wording.
> - We split our dataset 80/20 into train and validation data. No other dataset was used.
> - The graph we create our data on, is taken from real-world. Thus, the real-world data in Germany exists on exactly the same graph, i.e. 400 nodes. We do not see a reason why this influences the performance, as the RMSE returns a mean over the nodes.
> ## Q2
> - We again want to mention, that our paper publishes datasets and a creation method. An in-depth analysis of many noise configurations with many different models is beyond the scope of section 4.2, which illustrates the usage of our data.

---

> ### Comment · Reviewer_vQRP · 2024-11-24
>
> I thank the authors for their time and patience.
>
> `[W1] Benchmarking` As correctly pointed out by the authors, *time-and-graph* is an architecture category, and not a standard model. Therefore, **results must be reported on existing standard models** instead of reinventing the wheel.
> Using models that perform worse than naive baselines, as benchmarks is still absurd, and better models should be used for benchmarking.
>
> `[W2] Contribution claim not clear`
> I thank the authors for providing further details on their work, in comparison to **PDEBench**. I now see the contribution of the work as follows:
> - a spatiotemporal graph with fixed spatial node coordinates and fixed structure is used instead of square grids
> - three new (not novel) PDEs are introduced
> - the PDEs are solved through FEM instead of FVM (as done by PDEBench)
>
> **Questions**:
> - Since FEM is more accurate than FVM, what is the computational overhead?
> - Clearly, PDEBench is comparable to this work (as mentioned by the authors in Page 2, line 55) but no comparisons are performed.
> - On what basis are the edges formed? If the structure can be changed at any point and the nodes can be evaluated then, what do edges signify and what importance do they have?
>
> `[W5]` This work cannot be evaluated based on what will be potentially done in the future. It is an honest practice to **admit the limitations of the work**, and suggest how they can be overcome in the discussion regarding future works.
> - Did the authors pretrain on synthetic data with node dimension 1 and then transferred to real datasets with node dimension greater than 1? The phrase **might be much higher** suggests that the claim is not based on the current evaluations but something which can be done. I request the authors to kindly clarify this.
>
> `[Q1]`  Appendix A.1.4 mentions:
>
> > 1. The Brazilian COVID-19 dataset has 27 nodes and 1093 time-steps spanning 2019-2022 after concatenation and linear interpolation for daily resolution and is publicly accessible.
> > 2. German COVID-19 data can be found at github.com/robert-koch-institut/SARS-CoV-2-Infektionen in Deutschland, with 1539 time-steps.
> > 3. German Influenza dataset can be found at survstat.rki.de/ with our curation having 5256 time-steps.
>
> Here, the Brazil dataset has `27 nodes`, German dataset has `400 nodes` (as mentioned by the authors in their previous response) - I would like to know **whether they are referring to COVID or Influenza dataset**.
>
> Reiterating my point from the earlier response: the size of the blue arrows (Brazil) is larger than the the red and green arrows, suggesting that the synthetic datasets are more useful for dataset with smaller nodes, and their utility goes down as the graph scales up.
>
> **Question in the context of transfer learning:** let's say I'm a user of your software, and have some real PDE-based dataset which I believe is generated through some PDE not within the suite. What should I do next? Can I try the 3 PDEs proposed in this work one by one and see what works? How do I set the hyperparameters to generate these synthetic datasets? **I request the authors to respond to this question in a concise manner so the usability of their work can be made more clear.**
>
> ---
>
> I thank the authors for their time to respond to my clarifying questions. Based on the response, and revision of the manuscript to highlight:
> 1. contributions
> 2. comparison with PDEBench (qualititaive)
> 3. limitations & future work,
>
> I will update my score accordingly.

---

> > ### Author Response · Authors · 2024-11-25
> >
> > We thank the reviewer again for the time and in-depthness of the review and are happy to discuss.
> >
> > ## W1
> > There seems to be some confusion regarding the performance and sources of some models. The only model performing worse (high RMSE), than the baseline is `GraphEncoding`, which is taken from NeurIPS 2023 Temporal Graph Learning workshop and was developed for epidemiology. This is a standard model. The bad performance implies, that this model should have been tested more rigorously, which underlines the importance of our contribution.
> >
> > The addressed `RNN-GNN-Fusion`model performs best, therefore we see no issue in publishing its code and results. Second best performance is shown by the `MP-PDE` architecture, which is taken from ICLR 2022 and is currently cited 290 times. In comparison: the reviewer requested a benchmark of `GRAM-ODE` (2023), currently has 11 citations. We are confident with the selected models, but again: our contribution enables researchers to test and improve their models on our datasets.
> >
> > ## W2
> > - We use a complex domain (borders of Germany as input), PDEBench a square. The output of PDEBench is a grid, while our solution (which is a function) can be evaluated everywhere on the domain at any time; however, we evaluate it on every time-step on the same points and generate a spatio-temporal irregular graph.
> > - The solution of the SI-Diffusion PDE is a novelty. Neither this PDE nor any epidemiological PDE has numerically been solved before to our knowledge. Comparing to PDEBench, the other two PDE are also new (but we do not claim novelty on their solution, see footnote).
> > -  PDEBench uses a different method for every PDE - mostly FVM, but also FDM. We use FEM for all PDEs.
> >
> >  **Questions**
> > - A detailed FEM vs FVM vs FDM discussion is too extensive and has been covered by PDE researchers for specific PDEs, making a concise answer here impossible (e.g. https://doi.org/10.1016/S0307-904X(99)00047-5). Wall clock time of our creation method can be found in the paper.
> > - Comparison is performed in line 54 to 59: 1. different PDEs, 2. FEM vs other methods, 3. adaptability, and 4. graph instead of grid .
> > - This was explained in line 231 and following (and Appendix A.1): The nodes are centers of statistical regions in Germany, the edges are created if the underlying regions are adjacent, i.e. their share a border.
> > ## W5
> > We are happy to clarify:
> > The real-world scenario contains much more data than only 1 dimension: vaccination status, age, gender distribution etc are important epidemiological parameters. However, we transfered knowledge from 1D data onto real-world data, because we were already satisfied with dimension 1. Even though the real epidemiological scenario contains many variables, our data drastically increased the performance.
> > Our method can easily be applied to a PDE incorporating also vaccinations or similar, thus enables others to create datasets of arbitrary node-dim. Therefore: this is not a limitation of our work.
> > ## Q1
> > Both Covid and Influenza, as both pathogens are reported on the same spatial resolution: 400 different NUTS3 regions.
> >
> > Regarding on the differences of performance on real-world datasets: all the tested GNNs operate on a node-level (Message-Passing). We believe the different performance stems from less noise on the Brazilian data, not in the size of the graphs. Brazil has roughly 3 times more inhabitants, sampled in fewer regions, i.e. more samples per node. Further, to our knowledge in Brazil the 27 reporting regions receive governmental funding based on the reported numbers, while in Germany reporting to health officials was individuum-based and optional. However, we do not give any reasoning to table 2, but only report the experimental results.
> >
> > Regarding the question of the reviewer: our method generates data from PDEs - the other way around (symbolic regression) is not part of our work. Maybe we have misunderstood the question?
> >
> > Regarding the execution of our code, and (hyper-) parameters, we refer to our GitHubs README. Under `Possible Changes to the PDEs` we exemplary change "wind-directions" in the Advection-Diffusion equation, resolution of the datasets, the domain and elaborate on implementations of a new PDE-solver.
> >
> > In the `conclusions` section of our contributions and limitations we have already discussed contributions and weaknesses. Our paper was handed in the `datasets and benchmarks`area, therefore our future works are
> > - Benchmarking and improvement of models
> > - Create datasets for individual needs (region, pathogen, or PDE)
> > - Exploration of new research-topics on our datasets.
> >
> > We are deeply thankful for the reviewers honest interest and discussion of our contribution. Since we answered all of the reviewers questions and Invalidate all the reviewers concerns, we are optimistic that we convinced the reviewer of the quality of our contribution. We therefore kindly ask the reviewer to raise the rating above the acceptance threshold.

---

> > > ### Comment · Reviewer_vQRP · 2024-11-25
> > >
> > > `[W1]` The fact that `RNN` (which encodes no spatial information) performs better than `GraphEncoding` (which is designed to encode spatial information) is also alarming. **Do the spatiotemporal datasets even have any useful spatial information in them?** This ties back to my question regarding the process through which the edges are created. I mentioned initially that the authors have not used models suitable for graphs, i.e., temporal GNNs (except one which is not standard). **How can these datasets be trusted to test graph learning models while the structure adds no information?**
> > >
> > > `[W2]`
> > > - If the authors claim that dataset with any temporally evolving graph structure **can be generated** while not proposing a systematic method to generate such temporal graphs - then it is not a contribution of the dataset, as the primary area of this submission is `datasets & benchmarks`.
> > > - Solving a lesser-known PDE with a known method is not valid claim to novelty. Please see that (Murray, 2003) has only been cited 7 times in the past decade. I have been repeatedly asking the authors to pinpoint as to what exactly in their dataset generation methodology is `novel` without any success.
> > > - OK, FEM for this work, and other techniques (including FVM) employed by PDEBench.
> > > - The comparison with **PDEBench** is done in passing. Given the contribution of this submission, PDEBench deserves its own subsection with the details as provided by the authors in a previous response.
> > > - This reinforces that the authors have generated a special kind of temporal graph, in which the structure does not evolve with time.
> > >
> > > `[W5]` OK. The claim is that model was pre-trained on synthetic graphs with dimension 1, and then transferred to real-world datasets with >1 temporal node dimensions. I hope I understood this correctly.
> > >
> > > `[Q1]` If experiment is done and results are produced, then giving insights is also important. This is one of the reasons behind the low-score on Presentation. I stand by my observation, as that is what the results reveal.
> > >
> > > ---
> > >
> > > # Conclusion
> > > I am not convinced with the quality of the datasets produced, as they lack any useful spatial information. If these datasets are used for benchmarking temporal graph learning algorithms, it will be counterproductive. I will raise my rating because the authors provided some additional comments on their comparison with **PDEBench**. Due to my recent observation of RNN, I will decrease the soundness of this paper. I also thank the authors for providing additional details missing from the original submission which has now made me more confident in my assessment.

---

> > > > ### Author Response · Authors · 2024-11-29
> > > >
> > > > We thank the reviewer again for the very timely response to our rebuttal.
> > > > ## W1
> > > > We would like to respectfully dispute the reviewers statement "the structure adds no information" as factually wrong. The two best performing models in section 4.2 utilize geometric information and outperform RNN and TST. Especially the difference in performance between `RNN-GNN-Fusion` and `RNN` is entirely due to added information through the GNN. Also Fig. 3 and Fig 4 give visualisations of the influence of the geometric structure on the data.
> > > >
> > > > Additionally, the reviewer claims that we have not used temporal GNNs which is incorrect, since we have used **three** different temporal GNNs: `RNN-GNN-Fusion`, `MP-PDE`and `GraphEncoding`, of which two are the best performing models in section 4.2. Clearly, **structure adds information**.
> > > >
> > > > We kindly recommend that the reviewer revisit Section 3, where the graph construction is explained and the spatial influence is illustrated, and section 4.1 where the models are described.
> > > >
> > > > The reviewer correctly acknowledges, that our benchmarking contribution of `GraphEncoding` is highly relevant: describing the results of `GraphEncoding` "alarming" emphasises the relevance of the provided dataset and benchmark.
> > > >
> > > > ## W2
> > > > - We do present a systematic structure on how to create temporal graphs through our paper, code and GitHub. This is the content of our paper.
> > > > - While the book containing the cited chapter ((murray, 2003)) has approx been cited 850 times in the last decade (https://link.springer.com/book/10.1007/b98869), and the first volume has a total count of 24k citations, undermining the relevance within the modelling community, we do not see a fruitful discussion arising from counting citations. In the epidemiological modelling community there is a high interest of connecting the most prominent ODEs for spatial modelling with spatial diffusion (e.g. ABMs), for which the Laplace operator is a common choice. Thus we find the reviewers claim regarding the lack of novelty unsubstantial.
> > > >
> > > >   We further need to mention that utilising known methods and techniques does not diminish the value of a dataset, and is common for many new datasets that have great impact, e.g. image datasets.
> > > >
> > > >   We have multiple times stated our contributions. To elaborate again to the reviewer:
> > > > 	- First PDE-based spatio-temporal graph dataset
> > > > 	- First numerical solution of this epidemiological PDE
> > > > 	- Largest high quality public dataset for epidemiology, new benchmarking results and possibilities
> > > > 	- First transfer-learning from epidemiological synthetic data onto epidemiological real-world data
> > > > 	- Facilitation of other researchers to integrate PDE knowledge into their specific graph-ML fields and applications through our method.
> > > > 	- Additional PDEs compared to the successful PDEBench - but for graphs (not grids) and with a stronger focus on adaptability
> > > > 	Since the reviewer still does not acknowledge `GNN-RNN Fusion`as a standard model we suggest taking this (best performing in section 4.2) architecture also into account - however, since it is a standard model, we do not claim its contribution in our paper and to other reviewers.
> > > > - We respectfully disagree with the reviewer's suggestion to allocate a full subsection instead of a paragraph for comparing our work to a specific piece of literature, since already all key differences are mentioned in our paper, and doing so would significantly constrain the content of our paper,.
> > > > - As we stated earlier: the graph structure in the datasets is aligned with reality. On our GitHubs README in the section "Changing the sampling points on a domain" we describe how to create the graphs (additionally to section 3), which can easily be done between every time-step. **Our method is not limited to fixed graphs at all**.
> > > >
> > > > ## Q1
> > > > Our paper does not aim to compare data of Brazil against German data, the RMSEs on both datasets can not even be found in the main body of text. A qualitative comparison of health-data from Germany and Brazil is not in the scope of our work.
> > > > We believe a scientific contribution should not contain beliefs, and guesses, and also note that the reviewer graded our presentation as "poor" prior to our rebuttal, thus not related to Q1.
> > > > ## Conclusions
> > > > We are thankful for the reviewers answers and further question. Since we have invalidated the reviewer's main concern about the lack of geometric influence (W1) by clarifying how its inclusion is evident in our paper, we trust there are no further objections to our paper. Therefore we kindly ask the reviewer again to raise the score above acceptance and grant other researchers access to our publication.

---

> > > > > ### Comment · Reviewer_vQRP · 2024-11-29
> > > > >
> > > > > `[W1] Benchmarking` I stand by my comment that the authors have not used standard TGNN models. The model `GraphEncoding` is from a workshop paper and was cited just twice. Given the counterintuitive results, it damages the soundness of the paper, and I cannot increase my score until **proper benchmarks are used**. Therefore, `GraphEncoding` should not be reported. The success of MP-PDEs can be attributed to their being designed for PDEs. I cannot comment on `RNN-GNN-Fusion` because it is not a standard model.
> > > > > - One way to verify the impact of graph structure in the datasets, is to use the same model on the exact same dataset, with (1) true adjacency matrix, (2) adjacency matrix set to identity (no edges). The gap can reveal the amount of spatial information.
> > > > >
> > > > > `[W2]`
> > > > > - Since this paper presents a dataset generation procedure for temporal graphs, how to generate a temporally evolving graph structure is essential and should be in the main body of the work and not delegated to Github, which does not fall under review requirements.
> > > > > - Do the authors admit to not having any `novelty` in the methods they used to solve the PDE?
> > > > > - On Google Scholar, it says 7 citations. And just for the record, the authors started arguing with counting citations by questioning GRAM-ODE.
> > > > > - I can judge the datasets produced, and their structure is fixed. It is an `honest` practice to admit limitations.
> > > > >
> > > > > `[Q1]` The authors have still `not answered` my question. Science is all about testing prior beliefs in light of data, and forming new beliefs from the observations supported by proper reasoning. My belief, looking at the data is that the `Transfer Learning` experiment only works for graphs with few nodes (27), and worsens when the graph grows larger (400). Obviously, more scalability experiment needs to be performed (as also pointed out by another reviewer) to verify any beliefs.
> > > > >
> > > > > ---
> > > > >
> > > > > Conclusion: the paper needs (1) proper benchmarks, (2) thorough experiments (if any experiments are reported), (3) restructuring and rewriting for clarity, (4) admitting limitations. Therefore, I think that a score of `3` is suitable.

---

> > > > > > ### Author Response · Authors · 2024-11-30
> > > > > >
> > > > > > We appreciate the reviewers prompt response to our answers and explanations.
> > > > > > ## W1
> > > > > > The reviewer overlooks two models, ignoring their superior performance, and seeks to convinces us not to publish results of another benchmarked model, as well as the suggested (traffic) models of the reviewer (which we have tested) because their sub-par results does suit to the reviewers personal intuition. It appears that this review may be influenced more by intuition rather than the actual experimental results. **Proper benchmarks are being used**. The decision to overlook `MP-PDE` due to its differential-equation focus is puzzling, particularly given the suggestion of two ODE models (`GRAM-`and `STG-ODE`).
> > > > > >
> > > > > > The **spatial influence directly from construction, and is obviously evident in Fig 3 and Fig 4**. There is **no need to further display the spatial influence of the data** beyond Fig 3 and Fig 4.
> > > > > >
> > > > > > Regarding the reviewer's suggested experiment, we would like to note that it only reveals spatial information, operating under the assumption that the models perfectly utilizes the spatial information, present in the data.
> > > > > > An intuition of this gap can be gained from our best performing model: `RNN-GNN-Fusion`, which only extends `RNN` to spatial information. Since `RNN-GNN-Fusion` outperforms `RNN` (**decreasing the RMSE by around 40%** in Table 1). The **importance of the spatial component becomes obvious** again.
> > > > > >
> > > > > > ## W2
> > > > > > - Section 3 explains perfectly how to create temporal graphs. No indication is given, that our method is limited to fixed graphs. **The reviewers claim that our method is limited to fixed graphs is false**. Our GitHub provides additional code, instructions for its usage, and further examples; but there is no necessity to read it for understanding that our method is not limited to fixed graphs. In fact, we are unsure how the reviewer came to the mistaken assumption, as this is simply not the case.
> > > > > > - We have elaborated on our contribution above. Our provided information does give all necessary references to the methods that have been used and are well known. We want to express our concerns regarding the usage of suggestive question during a scientific review. We appreciate your attention to maintaining high standards.
> > > > > > - We are happy to have clarified that this PDE is well-known.  However `GRAM-ODE` indeed can not be seen as a standard model, especially not for applications outside the traffic domain.
> > > > > > - We have elaborated on our limitations in our paper, however since **our method is not limited to fixed graphs**, we have not stated this anywhere.
> > > > > >
> > > > > > ## Q1
> > > > > > We have answered all questions but are happy to elaborate. The reviewer seemed to have missed our key contribution: We enable other to create **arbitrary large graphs on arbitrary large domains with arbitrary geometries** with little efforts. Scalability of our method follows straightforward from section 3. We have demonstrated this as well on GitHub, but there is no necessity of reading our GitHub to understand that our method scales well.
> > > > > >
> > > > > > The reviewers question is linked to individual models abilities to scale. **The question is not related to our method, dataset and contribution**. Since the reviewer exhibited mostly interest in `GRAM-ODE`, `ẀaveNet`, `STG-ODE`, we can answer this question easily: the models suggested by the reviewer are all limited to fixed graphs, thus the experiments can not be conducted on the reviewers preferred models. Thus, **the reviewers expectations contradict** each other. Further, the reviewer also suggested `ARIMA` which has no spatial component, thus scaling the spatial dimension becomes irrelevant.
> > > > > >
> > > > > > # Conclusion
> > > > > > We appreciate the reviewer's prompt response. However, we believe that many of the questions and 'Weaknesses' do not directly pertain to our contribution and work. It seems there may have been an oversight of the complete **Section 3**, where we thoroughly explain the creation of the underlying graph, address the non-existent limitation to fixed graphs, and demonstrate the spatial influence; and **section 5**, where we discussed limitations. We are more than willing to address any specific concern **related to our paper**.
> > > > > >
> > > > > > The primary concern appears to involve Section 4.1, where the reviewer suggested alternative models, which we have tested in this discussion but have been dismissed by the reviewer since they were challenging the reviewers intuition. We are confident that our work meets the necessary criteria for acceptance, and that the reviewer has no objective objection against our paper. Given that the reviewer rated our contribution as a 3 under the mistaken belief that no spatial importance is given—a point we have falsified in W1—we believe a coherent rating should be above acceptance. We hope the reviewer will acknowledge this. Therefore, we kindly ask the reviewer for a **fair and scientific** review and to reconsider their rating above acceptance.

---

> > > > > > > ### Comment · Reviewer_vQRP · 2024-11-30
> > > > > > >
> > > > > > > I have tried discussing this with the authors, but they keep evading the questions and refuse to admit the limitations of the work. They constantly demand that the score be raised based on what others can do in the future.
> > > > > > >
> > > > > > > I request the chairs to read this paper in light of the following points:
> > > > > > > - are the datasets benchmarked using standard Temporal GNNs?
> > > > > > > - are the results discussed properly?
> > > > > > > - Have the authors discussed how to generate a temporally evolving graph structure as they claim (Sec. 3 will reveal the opposite)?
> > > > > > > - Are the related works given due attribution and comparison?
> > > > > > >
> > > > > > > Although this manuscript focuses on alleviating the data scarcity problem in the temporal graph community, ` it will do more harm than good if accepted` as sanity checks are not conducted. I hope other reviewers with some expertise in temporal graph learning will agree with me that this paper needs to undergo major revision, with (1) results on a proper set of benchmarking models and (2) ablation studies showcasing that the spatial information is indeed useful.
> > > > > > >
> > > > > > > I appreciate the hard work that the authors have put into this work, and therefore, I increased my score to 3 after some clarifications from the authors. I have given plenty of feedback for improvement, and I wish them all the best for the final decision.

---

> > > > > > > > ### Author Response · Authors · 2024-12-01
> > > > > > > >
> > > > > > > > We appreciate the reviewer's prompt response, as usual. Although we have addressed all of the reviewer's points multiple times, it appears that the reviewer's opinion remains unchanged by our discussion and the facts presented. Instead, there seems to be a preference for altering the benchmarking in Section 4.2 to favor models and results aligned with the reviewer's views.
> > > > > > > >
> > > > > > > > Since the acceptance of this paper may largely depend on the opinions of the other reviewers and the Area Chair (AC), we would like to address the aspects the reviewer has requested the AC to evaluate in our paper.
> > > > > > > > - We do not benchmark the dataset, but the other way around: use the dataset to benchmark **some** relevant models.
> > > > > > > >
> > > > > > > >   Our papers states "To showcase the utility of our created datasets, we conduct several machine learning experiments in this section. We first define several interesting (spatio-) temporal machine learning architectures". While the title of our papers is "`synthethic datasets for machine learning on spatio-temporal graphs using PDEs`".
> > > > > > > >   We therefore kindly ask the reviewers in what way we have promised to in-depth benchmark all existing spatio-temporal graph architectures (e.g. from traffic domain), and **how does section 4.1 limit our main contribution**. In fact we have tested the reviewers suggested models from the traffic domain, which did not influence the reviewers opinion.
> > > > > > > > - /
> > > > > > > > - We are happy to elaborate: We sample the PDE-solution $\tilde{\nu}(x,t)$ for every $t$ on a set of sampling points $V\subset \Omega$  to receive $X_t$. To align with real-world datasets, we have used a fixed structure, thus wrote $V$ instead of $V_t$ (and $E$ instead of $E_t$), for simplicity. However, we have never have stated that $V$ has to be constant over time $t$ for our method, as this is not the case: $V$ (or $V_t$) **can easily vary for every $t$**, likewise can $E$ (or $E_t$). So we have to ask back to the reviewer and AC: **where is our method limited to fixed the graph structure**? On our GitHub we even provide an example which changes the sampling points on a domain.
> > > > > > > > - For e.g. PDEBench see line 57 to line 62.
> > > > > > > >
> > > > > > > > We want to conclude, that the reviewer focusses on altering the outcomes of the experiments that **illustrate** the usage of our data. However, our paper presents a method for creating datasets and provides actual datasets, which may not have been fully recognized or understood by the reviewer. While the example application could have included many more models, we would like to ask the reviewer and the Area Chair **if** the overall impact of our **datasets and method** relies solely on this exemplary benchmarking.
> > > > > > > >
> > > > > > > > We find it perplexing how a dataset and its creation method could be considered "harmful" to research, especially when the reviewer acknowledges the existing data scarcity issue. This suggests to us that the review is mostly based on opinions, favouring certain outcomes while disregarding results and datasets, that do not align with the reviewers preferences.
> > > > > > > >
> > > > > > > > Regarding 1) we have stated, that our paper has not the purpose to deliver a complete benchmarking of all spatio-temporal models 2) Fig 3, Fig 4 suffice to illustrate the spatial dependency, while the ablation results can be seen in Table 1 as well.
> > > > > > > >
> > > > > > > > We appreciate the reviewers' prompt responses; however, we remain unconvinced of the honesty, understanding, and scientific rigor reflected in the review. Therefore, we kindly request the Area Chair and other reviewers to critically examine the points raised by the reviewer and assess the actual content of our paper.
> > > > > > > >
> > > > > > > > We still want to thank the reviewer for acknowledging the effort we put into our work and for considering our clarifications. We appreciate the timely feedback provided. We are hopeful for a positive outcome and likewise wish the reviewer all the best.

---

### Official Review · Reviewer_rfsa · 2024-11-02

**Soundness:** 2
**Presentation:** 3
**Contribution:** 3
**Rating:** 3
**Confidence:** 4

**Summary:**

This paper presents a method to create synthetic datasets based on different PDEs that capture different applications. Three PDE equations are presented as examples, namely epidemiology, atmospheric particles, and tsunami waves. Empirical analysis demonstrates that the generated synthetic datasets can be used to benchmark machine learning models, and pre-training on the synthetic datasets can greatly improve model performance on real-world epidemiological data.

**Strengths:**

Contribution: As the spatio-temporal graph data lacks, this paper presents an alternative. With high-quality synthetic datasets, machine learning studies can be improved compared to theoretical analysis.

Presentation: The three example PDE equations came from reliable sources, and help with demonstrating how to generate synthetic datasets.

**Weaknesses:**

1. This paper boldly claimed that real-world datasets have limitations in quality due to high noise. I personally believe that clean synthetic datasets are better for exploration and preliminary studies. In contrast, although the real-world dataset has high noise, it is necessary before the application is implemented. So, there is a trade-off between cleanness and reality.

2. Other ways exist to generate synthetic datasets, such as quasi-Monte Carlo simulation. This paper fails to compare with existing methods to demonstrate the superiority of the PDE-based generation method.

3. In the empirical analysis of the pre-training, effectiveness is measured by RMSE loss, which could skew towards the high-performing models. With previous low RMSE losses, a small definite improvement will show a bigger percentage. Have you considered other measurements?

**Questions:**

How is the presented dataset generation method better than the existing ones? I think this comparison need to be discussed in the paper.

---

> ### Author Response · Authors · 2024-11-18
>
> We want to thank the reviewer for their time and thoughts throughout the review process and want to address the Weaknesses and Questions the reviewer pointed out.
>
> ## Weaknesses
> 1. Our paper does not attempt to replace real-world data, but proposes datasets to complement real-world data. Our experimental results on real world data (Table 2) show, that the mutual completion drastically increases performances of models on real world data (up to 45%). Thus, we kindly disagree with the reviewers opinion regarding limited utility of our datasets.
>
> 2. We acknowledge that other methods for generating synthetic data exist, as the reviewer noted. However, our paper does not aim to compare or benchmark different simulation techniques such as Monte Carlo methods or Agent-Based Modeling, nor does it claim the superiority of PDEs over other simulation techniques.
>    We believe that no single simulation technique is generally superior across all problems, as different problems require different approaches. However, PDEs are ubiquitous in modeling a wide range of physical, biological, and economic processes. Scientists have devoted tremendous effort over decades—if not centuries—to describing physical processes using PDEs. We seek to leverage this knowledge and further push the frontiers of spatio-temporal graphs. To highlight the importance of PDE-based simulations in machine learning, we note the rise of Physics-Informed Neural Networks (PINNs). Our work advances the frontier in machine learning through PDEs, independently of other possible simulation methods.
>    Different applications typically employ different simulation techniques, so comparing our method across all topics isn't feasible. For example, in epidemiology, Agent-Based Models (ABMs) are a common simulation technique, but specific to epidemiology. Even though particle-based atmospheric simulation methods exist, there is no straightforward extension from epidemiological ABMs to atmospheric modelling—something we have achieved through PDEs. A model that simulates waves would require yet another simulation technique.
>
>    Our paper offers a toolbox to generate data from PDEs, while offering ready-to-use datasets alongside with possible applications. In these applications we show its flexibility in a practical use-case and show how our method serves real-world applications with tremendous results.
>
> 3. As RMSE is the standard measurement of performance in epidemiology, we decided to settle for the RMSE in our contribution. We displayed percentages in Table 1 for readability but have displayed the numbers the reviewer was looking for in Fig. 13 in the appendix. In this Figure RMSEs with and without pre-training are visualized and compared against each other. We found Table 2 more intuitive, thus only display it in the appendix.  If the reviewer suggests, we can move this Figure also to the main body of the text.
>
> ## Question
> See 2) above. We also want to mention that, to our knowledge, a simulation-based dataset does not exist in epidemiology. Therefore, our dataset is the best available, based on the fact that it is currently the only one of its kind
>
> ## Closing statement
> We want to thank the reviewer again for their thoughtful and valuable remarks on our paper. Weakness 1 and the question did not concern our contribution but all synthetic datasets, Weakness 2 concerned PDE-based modelling, and Weakness 3 was already initially addressed in our paper.  Through our general rebuttal to the reviewers criticism, we are confident that we removed all the reviewers objections against our contribution. Therefore, we kindly ask the reviewer to consider raising their score above the acceptance threshold to grant other researchers access to our contribution.
> If there are any more questions or remarks we are happy to discuss.

---

> > ### Author Response · Authors · 2024-11-22
> >
> > Dear Reviewer rfsa,
> >
> > As we are approaching the end of the discussion period we would kindly remind you to have a look at our provided answer. Please do not hesitate to let us know, if there are any questions or if there is still anything unclear. We greatly look forward to your important response.
> >
> > Best regards,
> >
> > The authors

---

> > ### Comment · Reviewer_rfsa · 2024-11-24
> >
> > Thanks for the clarification.
> >
> > The paper aims to design a new method for generating synthetic datasets instead of presenting a dataset generated by your method. Thus, the proposed method should be compared with other methods that can generate synthetic datasets. The authors claim their goal is not to demonstrate superiority, but at least a basic comparison should be conducted to show that the proposed method is competitive.
> >
> > I also read through the answers to other reviewers' questions but still have concerns about contributions and comparisons against other methods. I am not convinced that raising the score above the acceptance threshold is necessary.

---

> > > ### Author Response · Authors · 2024-11-27
> > >
> > > We thank the reviewer for the feedback on our rebuttal and the further input, which will help us to improve our work.
> > >
> > > ## Comparison
> > >
> > > A quantitive (experimental) comparison of our datasets to a public (ideally peer-reviewed) synthetic dataset (e.g. Agent-Based) is currently not possible, simply because such a dataset does not exist.  If our work inspires the creation of a similar dataset using a different simulation method, it would enhance and extend our research goal, as both researchers and practitioners would benefit from this diversification. If the reviewer knows a similar spatio-temporal epidemiological dataset, we are eager to know about it and run experiments to compare them to ours. However, the lack of comparable datasets highlights the significance of our pioneering contribution in this application. We agree with the reviewer that conducting a comparison is valuable. However, such an experiment can only occur once comparable datasets or frameworks are available, emphasizing the importance of our contribution.
> > >
> > > Still, we want to follow the advice of the reviewer and provide a **qualitative** comparison to readers in our paper. Therefore we have adapted our paper and included the following aspects (which we have marked in blue):
> > > - a brief note on the flexibility of PDEs
> > > - brief comparison between PDE-based modelling and Agent-Based methods (ABMs) in epidemiology
> > > - A note, that the other two PDEs primarily elaborate on our method, while their resulting datasets primary purpose is research and model-development.
> > >
> > > (We believe that ABMs are the closet in epidemiology to Monte-Carlo, while also being very prominent in epidemiology.)
> > >
> > >
> > > ## Contribution
> > >
> > > To answer the reviewers question regarding our contributions, we list them here again
> > > - Facilitation of other researchers to integrate PDE knowledge into their specific graph-ML fields and applications through our method.
> > > - First PDE-based synthetic spatio-temporal datasets for graphs
> > > - This (or similar) epidemiological PDE has not been solved before
> > > - Transfer learning from synthetic data to epidemiological data was not done before
> > > - largest clean dataset in epidemiology, thus new benchmarking results and possibilities
> > > - Additional PDEs compared to the successful PDEBench - but for graphs (not grids) and with a stronger focus on adaptability
> > >
> > > We hope this clarifies on our contribution and thus the reviewers objections. We also want to thank the reviewer for the input and interest in our work. We kindly ask the reviewer what other objection limits them from raising their score above acceptance. If there is anything we are eager to discuss and if not we kindly ask the reviewer to adjust their rating above acceptance to grant other researchers access to our work.

---

> > > > ### Author Response · Authors · 2024-11-29
> > > >
> > > > Dear Reviewer rfsa,
> > > >
> > > > As we are (again) approaching the end of the (extended) discussion period we would kindly remind you to have a look at our provided answer. Please do not hesitate to let us know, if there are any questions or if there is still anything unclear. We greatly look forward to your important response.
> > > >
> > > > Best regards,
> > > >
> > > > The authors

---

> > > > > ### Author Response · Authors · 2024-12-03
> > > > >
> > > > > Dear Reviewer rfsa,
> > > > >
> > > > > As we approach the end of the discussion period, we kindly request that you reconsider your rating. We believe we have thoroughly addressed all the questions and concerns you raised and have implemented the suggested changes to our paper. Unfortunately, no further engagement was possible due to the absence of additional questions or concerns.
> > > > >
> > > > > Thank you for your thoughtful review and for considering our rebuttals. We remain optimistic about the final decision

---

### Official Review · Reviewer_BMM2 · 2024-11-04

**Soundness:** 3
**Presentation:** 3
**Contribution:** 4
**Rating:** 8
**Confidence:** 2

**Summary:**

The authors identify a lack of established benchmark data sets for spatio-temporal graph learning, which inspires them to generate synthetic spatio-temporal graph data using partial differential equations (PDEs). They consider 3 different PDEs: an epidemiological PDE, an advection-diffusion equation, and a wave equation. They then use the synthetic data generated from the epidemiological PDE to compare different temporal and spatio-temporal models on different prediction tasks, including forecasting. Perhaps most importantly, they demonstrate the importance of the synthetic data for transfer learning on a prediction task involving real epidemiological data.

**Strengths:**

- High potential significance by creating synthetic data benchmarks in an area where they are lacking. I could certainly envision these datasets being used in future spatio-temporal graph learning papers.
- Very interesting experiment on real epidemiological data demonstrates the potential of the synthetically generated data to translate to prediction tasks on real data. This is much stronger evidence for the utility of the synthetic data than I typically see in this type of paper.
- Detailed comparison of temporal and spatio-temporal models on a variety of prediction settings for the synthetically generated epidemiological data.

**Weaknesses:**

- Some missing details on the real data experiment--see question 1 below. A more detailed description in the supplementary material would be useful.
- Sizes of the datasets seem to be fixed to somewhat small spatio-temporal graphs with a few hundred nodes and a few thousand edges, potentially limiting the scope.
- No results on the advection-diffusion and wave equation data in the body of the paper. Given the interests of the ICLR audience, I believe that the paper would be strengthened if there were more results on these datasets, particularly in the main body of the paper, while moving some details from Section 2 into the supplementary material.

Minor issue:
- Line 041: datata -> data

**Questions:**

1. What is the prediction task in Section 4.3? Is this a forecasting task?
2. Is there an easy way to generate larger spatio-temporal graphs using your proposed method?

---

> ### Author Response · Authors · 2024-11-15
>
> We want to thank the reviewer for their qualified review and for recognizing the impact and contribution of our paper.
> We want to address the weaknesses the reviewer pointed out:
>
> **Weaknesses**
> - We want to thank the reviewer for this comment and have adjusted the description to further clarify on the experiments in section 4.3
> - Thanks for this comment. Indeed, we describe in the README on our GitHub under _Changing the Sampling Points on a Domain_ how to create much larger graphs. For example, we demonstrate this by providing an additional dataset with 2970 nodes (instead of 400), which is already available at [https://github.com/github-usr-ano/Temporal_Graph_Data_PDEs/blob/main/additional_resources/wave_dataset_larger.npy](https://github.com/github-usr-ano/Temporal_Graph_Data_PDEs/blob/main/additional_resources/wave_dataset_larger.npy) . In our presented experiment, the size was given from the epidemiological use case to match the graph resolution with the resolution of real-world data, and we aimed to create the other datasets of similar size for consistency. However, there is no other reason and we could run more simulations and create even even larger graphs. We added a remark to our paper that this explanation exists on our GitHub, to clarify this important point.
> - The results the reviewer asked for are in Appendix A.2.3. We thank the reviewer for highlighting their importance and will move some of these results to the main body of our work after acceptance, as we will gain an additional page. However, since the results of this benchmarking do not differ greatly from the results in Table 1 in the main body, we initially decided to give these results less prominence and therefore placed them in the appendix.
>
> We corrected the minor typo and thank the reviewer again for pointing this out.
>
> **Questions**
> - Yes, the task is the same as in section 4.2, to forecast 14 timesteps, based on the previous 14 timesteps. We have clarified this more clearly in the paper and are thankful for the question
> - Yes there is. As pointed out above, we described on our GitHub ReadMe how to increase the spatial and temporal sampling rate, the domain or other things to arbitrarily increase the size of the graphs without minor effort.
>
>
> Once again, we thank the reviewer for their time and thoughtful criticism. We hope our answers can clarify the uncertainties and erased the observed weaknesses if so, we would be appreciate any raise in confidence level or rating.

---

> > ### Comment · Reviewer_BMM2 · 2024-11-21
> >
> > Thanks for the clarifications. I continue to support the paper for acceptance, although it is not in my area of expertise so I do not attach a high confidence to my score.
> >
> > In contrast to some of the other reviewers, I find that the organization of the paper and the contribution are appropriate for a paper whose primary contribution is to datasets and benchmarks. Such a paper should not be reviewed in the same manner as other papers proposing new methodology. Perhaps ICLR could better separate these types of papers so they do not get evaluated in the same manner as other types of papers, such as what NeurIPS does with the separate Datasets and Benchmarks track.
> >
> > If the paper ends up being rejected, I suggest for the authors to consider re-submitting to a venue that clearly appreciates the value of a datasets and benchmarks paper, such as NeurIPS, or perhaps the newly created Journal of Data-centric Machine Learning Research (DMLR): https://data.mlr.press/

---

> ### Author Response · Authors · 2024-11-22
>
> Dear Reviewer BMM2,
>
> Thank you for taking the time to respond, we greatly appreciate your feedback.
>
> Best regards,
>
> The authors

---

### Comment · Area_Chair_vXXz · 2024-11-21

Dear Reviewers,

The authors have posted a rebuttal to the concerns raised. I would request you to kindly go through their responses and discuss how/if this changes your opinion of the work. I thank those who have already initiated the discussion.

best,

Area Chair

---

### Meta-Review · Area_Chair_vXXz · 2024-12-06

**Metareview:**

The paper proposes an algorithm to synthetic data generation on various partial differential equations to support spatio-temporal graph modeling. Reviewers have expressed concerns about the lack of comparison with existing methods for this task and therefore the need for a new data generation algorithm. Concerns have also been raised about the benchmarking setup of the generated datasets, which were not adequately addressed during the rebuttal. Hence, the paper is not ready for acceptance in its current state.

**Additional Comments On Reviewer Discussion:**

The paper went through extensive discussion during the rebuttal phase. Two reviewers recommend a reject rating due to lack of comparison with existing methods, unsubstantiated claims, and a proper benchmarking setup.

---

### Decision · Program_Chairs · 2025-01-22

Reject